# Epistemic Uncertainty Quantification To Improve Decisions From Black-Box Models

**Sébastien Melo[1], Gaël Varoquaux[1,2] & Marine le Morvan[1]**
[1]Soda team, Inria Saclay, Palaiseau, France
[2]:probabl., Paris, France
`{sebastien.melo,gael.varoquaux,marine.le-morvan}@inria.fr`

## Abstract

Distinguishing epistemic uncertainty (model ignorance) from aleatoric uncertainty (task randomness) is critical for reliable AI systems, yet standard confidence evaluation metrics capture different and incomplete aspects of uncertainty. While AUC and accuracy measure predictive signal, proper scoring rules assess overall uncertainty, and calibration metrics isolate part of the epistemic uncertainty but ignore within-bin heterogeneity of errors, known as grouping loss. We bridge this evaluation gap by introducing asymptotically consistent and sample-efficient estimators of the grouping loss and excess decision risk, providing a fine-grained assessment of epistemic uncertainty that complements existing calibration metrics. Applied to LLM question-answering with inherent aleatoric noise, our estimators reveal substantial grouping loss which decreases with model scale and instruction tuning. Their local nature enables automatic identification of subgroups with systematic over- or under-confidence, supporting interpretable confidence audits. Finally, we leverage these estimates to design LLM cascades that defer high excess decision risk predictions to stronger models, achieving higher accuracy at lower cost than competing approaches.

## 1 Introduction: a need to qualify prediction uncertainty

The recent European Artificial Intelligence (AI) Act requires careful oversight in some AI applications (EUC, 2024). In high-risk settings, predictive uncertainty behind a model's classification choice is central to proper risk assessment and flexible deployment (e.g. adapting to changing regulation on when the risk is considered sufficiently low). For example, some hospitals use models to estimate a patient's likelihood of surgical success based on their medical records (Mahajan et al., 2023), but these use cases call for thorough evaluation (Senge et al., 2014).

In many high-stakes settings, the best answer is not clear-cut. For instance, in the case of surgery prognosis, many unforeseen events can alter the outcome. A question such as "Will this patient relapse within the next 6 months?" must be answered with a confidence score that accounts for the amount of information available on the patient, unlike a question such as "Was Einstein born before 1800?", which admits a deterministic answer. The corresponding challenge of *aleatoric uncertainty* is often overlooked when evaluating LLMs confidence scores (Xia et al., 2025), which is at odds with their usage, e.g. in health settings (Jiang et al., 2023; Singhal et al., 2023; Omar et al., 2025; Katz et al., 2024).

Confidence scores may arise naturally from the continuous function that powers many classifiers, such as the simple logistic regression, or neural network with softmax output activation. But other settings require confidence score elicitation, as with LLMs. Indeed, LLMs output natural language answers which require dedicated techniques to be associated to a confidence (Shorinwa et al., 2025; Vashurin et al., 2025). Our work does not focus on this confidence scores elicitation, but rather on their evaluation. Evaluation and elicitation are often conflated, as an evaluation gives a signal that can often be used to improve confidence scores in return (e.g. measuring calibration error is an evaluation, but recalibration is an improvement technique). And yet, to decide whether to trust an AI system's answer, or whether to deploy it, in particular in a high-stake settings, the question of how good is its confidence score is crucial.

But how to measure the "optimality" of confidence scores? Confidence scores should capture the probability that the predicted outcome does occur. Standard metrics such as accuracy or AUC fail to capture this probabilistic aspect, as they cannot detect over- or under-confidence. Calibration (Gneiting et al., 2007) addresses this by comparing confidence scores to the rate of observed outcome. However, it is a control on average: a classifier may be well-calibrated overall yet overconfident on some subgroups and underconfident on others, with the two effects canceling out (Perez-Lebel et al., 2023). An optimal confidence score should reflect the true probability of the outcome depending on characteristics of the input $X$, not just match observed frequencies on average. For an input with insufficient information (e.g. not enough details in an LLM query), even the best predictor comes with a lot of uncertainty, called "aleatoric uncertainty". It is then a virtue of an AI system to report it (Kendall & Gal, 2017), as predictions are bound to come with errors and reporting this uncertainty can lead to better decisions. On the other hand, the errors of an AI system may arise from imperfections in this system, not using well the available information. Such uncertainty is known as epistemic uncertainty. A well-designed confidence score conveys both: expressing uncertainty when information is insufficient and indicating where the model could, in principle, improve.

Measuring the epistemic uncertainty, i.e. whether a model is knowledgeable enough, is highly desirable because it indicates where a model can be improved, either globally or locally (Wimmer et al., 2023). Global improvements might involve fine-tuning with additional data, while local improvements could include deferring decisions to a human expert or a more specialized, but costlier, model. Such deferral is inefficient if the uncertainty is aleatoric, since in that case the prediction is already optimal given the available input. Teasing out epistemic from aleatoric uncertainty is a very difficult task and arguably the holy grail of learning theory. Some metrics –"proper scoring rules" (Gneiting & Raftery, 2007), such as Brier score or log loss– can select models with smallest epistemic error; but they do not quantify the remaining epistemic error. For a calibrated model, residual epistemic error arises from heterogeneity in the error: the model may be overconfident on some inputs and underconfident on others (see Figure 1), a phenomena captured by the grouping loss (Kull & Flach, 2015; Perez-Lebel et al., 2023).

We tackle epistemic error quantification, to characterize how suboptimal a model is. We contribute:

- **Sample-efficient and local epistemic uncertainty estimators**: We introduce asymptotically consistent estimators for grouping loss and excess decision cost that bypass the limitations of traditional confidence binning.

- **Auditing LLM reliability**: We demonstrate that LLMs suffer from significant grouping loss that decreases with model scale and instruction tuning. By providing local estimates, our approach also enables fine-grained and interpretable confidence audits, identifying specific data subgroups where models exhibit systematic over- or under-confidence (see Figure 1).

- **LLM decision deferral**: We further show that our per-sample excess decision cost estimates enable superior LLM cascades, outperforming standard predictive routing by triggering deferral only when the epistemic risk is high.

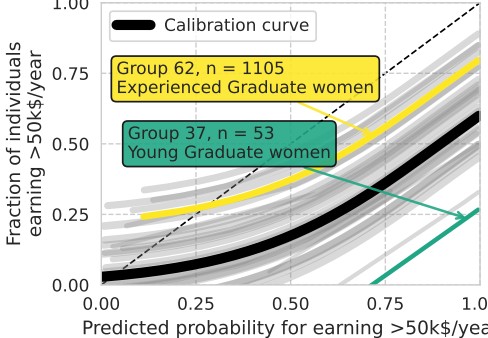

Figure 1: **Over and under-confident groups for Llama 3.3 Instruction-tuned 70B** on the ACSIncome dataset. Groups were obtained by applying our estimator to the confidence scores provided by the LLM. Exact group definitions obtained from the tree cuts are:
Group 37 (green): Master's degree and higher, Age $\leq 24$, Female.
Group 62 (yellow): Professional degree beyond a bachelor's degree, Age $\geq 37$, Female.

## 2 BACKGROUND AND RELATED WORK

**Notations** We consider a binary classification task where $(X, Y) \in \mathcal{X} \times \{0, 1\}$ are jointly distributed random variables representing the input space and class labels respectively. We write $f^*(X) \stackrel{\text{def}}{=} \mathbb{P}[Y = 1 \mid X]$ the *unknown* posterior class probabilities, and $f : \mathcal{X} \mapsto [0, 1]$ the confidence scores.

### 2.1 CALIBRATION

Calibration (Lichtenstein et al., 1977) measures whether the confidence scores produced by a model $f : \mathcal{X} \to [0, 1]$ match, *on average*, the observed frequency of positive outcomes.

**Definition 1** (Calibration). *A probabilistic classifier $f$ is calibrated if*

$$\forall p \in [0, 1], \quad \mathbb{E}[Y \mid f(X) = p] = p.$$

We note $c \circ f(X) \stackrel{\text{def}}{=} \mathbb{E}[Y \mid f(X)]$ the *calibrated score* ($c$ is a *recalibration* function). The most widely used calibration metric is the Expected Calibration Error (ECE) (Naeini et al., 2015), which measures the difference between predicted confidence and observed frequency by averaging absolute differences across probability bins. Cruz et al. (2024) is among the few studies to quantify LLM uncertainty on tasks with inherent aleatoric uncertainty using the ECE. They found that base LLMs tend to be underconfident, while instruction-tuned models often become overconfident.

### 2.2 PROPER SCORING RULES AND GROUPING LOSS

*Proper scoring rules* evaluate a predicted probability distribution $p$ against an observed outcome $Y$ (Dawid, 1986; Gneiting & Raftery, 2007). They can be decomposed to separate epistemic uncertainty (due to the model's lack of knowledge) and aleatoric uncertainty (due to inherent data randomness), while also isolating a calibration component.

**Theorem 1** (Proper scoring rule decomposition (Kull & Flach, 2015)). *Let $\phi$ be a proper scoring rule and $d_\phi$ its divergence. Then the expected loss writes as:*

$$\underbrace{\mathbb{E}[d_\phi(f(X), Y)]}_{\text{Expected Loss}} = \overbrace{\underbrace{\mathbb{E}[d_\phi(f(X), c \circ f(X))]}_{\text{Calibration Loss (CL)}} + \underbrace{\mathbb{E}[d_\phi(c \circ f(X), f^*(X))]}_{\text{Grouping Loss (GL)}}}^{\text{Epistemic loss (EL)}} + \underbrace{\mathbb{E}[d_\phi(f^*(X), Y)]}_{\text{Aleatoric Loss}}. \quad (1)$$

The above shows that the epistemic loss can be further decomposed into calibration and grouping components, highlighting that calibration only provides a partial view of epistemic error. When $\phi$ is the Brier score, the *grouping loss* corresponds to a variance term.

**Definition 2** (Grouping Loss (Lemma 4.1 Perez-Lebel et al., 2023)). *If the proper scoring rule $\phi$ is the Brier score, the grouping loss is given by:*

$$\text{GL} \stackrel{\text{def}}{=} \mathbb{E}\left[(f^*(X) - c \circ f(X))^2\right] = \mathbb{E}_p\left[\mathbb{V}[f^*(X) \mid f(X) = p]\right]. \quad (2)$$

The grouping loss measures the variance of true probabilities within a level set of the model $f$, reflecting that the model groups together inputs with different true probabilities. Modern neural networks were shown to exhibit grouping loss (Perez-Lebel et al., 2023; Chen et al., 2024), especially under distribution shift. In the remainder of this paper, we use the Brier score as proper scoring rule.

### 2.3 EXCESS DECISION RISK

A key distinction lies between assessing probabilistic suboptimality—the discrepancy between scores and true probabilities—and decision risk, which captures the *operational* suboptimality of the resulting actions. The former is typically measured by proper scoring rules, while the latter evaluates the decision rule itself.

Confidence scores are often used to make decisions, i.e., mapping a sample $X$ with confidence $f(X)$ to an output $Y \in \{0, 1\}$. Given a cost matrix $\Lambda \in \mathbb{R}^{2 \times 2}$ where $\Lambda_{i,j}$ is the cost of predicting class $i$

when the true class is $Y = j$, decision theory (Elkan, 2001) gives the optimal decision rule from $f^*$:

$$\delta^* \stackrel{def}{=} x \mapsto \mathbf{1}_{f^*(x) \geq t^*} \qquad \text{with} \quad t^* \stackrel{def}{=} \frac{\Lambda_{1,0} - \Lambda_{0,0}}{\Lambda_\Delta}; \quad \Lambda_\Delta \stackrel{def}{=} \Lambda_{1,0} + \Lambda_{0,1} - \Lambda_{0,0} - \Lambda_{1,1}.$$

$t^*$ is the optimal threshold that depends on misclassification costs. Given a model $f$ and a selected decision threshold $t$, we consider the decision rule $\delta_{f,t} \stackrel{def}{=} x \mapsto \mathbf{1}_{f(x) \geq t}$ and the associated expected cost of the decision $EC(\delta_{f,t}, x) \stackrel{def}{=} \mathbb{E}_{Y|X}[\Lambda_{\delta_{f,t}(X),Y} \mid X = x]$. Suboptimal confidence scores lead to suboptimal decisions, resulting in excess decision risk. In particular, Perez-Lebel et al. (2025) showed that the excess decision cost can be decomposed into two components: one arising from calibration errors in the confidence scores, and one due to grouping errors.

**Proposition 1** (Excess decision risk decomposition (Perez-Lebel et al., 2025)). *Let $\Lambda \in \mathbb{R}^{2 \times 2}$ be a cost matrix. The excess decision risk $\mathcal{R}_f$ can be decomposed as:*

$$\mathcal{R}_f \stackrel{def}{=} \mathbb{E}[\mathcal{R}_f(x)] = \underbrace{\mathbb{E}[EC(\delta_{f,t}, x) - EC(\delta_{c \circ f, t^*}, x)]}_{\mathcal{R}_f^{CL}} + \underbrace{\mathbb{E}[EC(\delta_{c \circ f, t^*}, x) - EC(\delta_{f^*, t^*}, x)]}_{\mathcal{R}_f^{GL}} \quad (3)$$

*where $\mathcal{R}_f^{CL}$ and $\mathcal{R}_f^{GL}$ are the excess costs stemming from miscalibration and grouping loss.*

As for the evaluation of confidence scores, calibration alone does not fully capture the suboptimality of an *individual* decision. Perez-Lebel et al. (2025, prop 3.2) proposed calibration risk and grouping risk estimates averaged *per level set*. Specifically, the bin-wise calibration risk is defined as $\mathbb{E}[\mathcal{R}_f^{CL}(X)|f(X) = f(x)]$ and admits a closed-formed expression recalled in Appendix B.1.

## 2.4 RELATED WORK

**Epistemic Uncertainty Estimation** Bayesian-inspired methods like MC-Dropout and Deep Ensembles are common techniques for this purpose (Gal & Ghahramani, 2016; Lakshminarayanan et al., 2017). However, a key criticism is that they primarily capture approximation uncertainty (due to a finite dataset) instead of model uncertainty (stemming from a misspecified model class), even though both contribute to epistemic uncertainty (Hüllermeier & Waegeman, 2021; Lahlou et al., 2023). Bickford Smith et al. (2025) further criticizes the many existing definitions of epistemic uncertainty that are derived solely from the learned distribution $f$ (e.g., MC-Dropout and Deep Ensembles) and advocates for the use of externally grounded epistemic evaluation (e.g. using ground truth realizations of the output). In this setting, Mimori et al. (2021) proposes an epistemic loss estimator that leverages multiple labels per sample, while Johnson et al. (2024) introduces a training paradigm where a model learns to predict pairs of outcomes for a given sample to estimate epistemic uncertainty. However, multiple annotations are not always available. Finally, Perez-Lebel et al. (2023) propose a lower-bound estimation of the grouping loss using local averages within bins of equal predicted probability.

**Calibration for Decision Making and Net Benefit** In the medical decision-making literature, expected decision risk minimisation is known under the name of Net Benefit maximisation, with the optimization performed over the decision threshold (Rousson & Zumbrunn, 2011). Interestingly, selecting the threshold that minimizes the expected decision risk is equivalent to calibrating the candidate model for use with the optimal threshold $t^*$ (Proposition 3.1 in Perez-Lebel et al., 2025; Van Calster & Vickers, 2014). Pfohl et al. (2022) introduced algorithmic fairness objectives in the Net Benefit framework, showing that different subgroups may require different decision thresholds. Indeed, optimizing decisions on average from calibrated scores $c \circ f(X)$ (as with net benefit) does not account for group- or individual-level disparities.

**Decision Deferral** Decision deferral studies strategies by which a model abstains when uncertain, improving reliability in high-stakes domains (Hendrickx et al., 2024). Simple approaches defer based on a confidence or uncertainty threshold, but these assume that the model's confidence scores are reliable (Wang et al., 2022; Jitkrittum et al., 2023). More sophisticated methods learn a deferral policy by training a dedicated rejection function alongside the classifier (Mozannar & Sontag, 2021; Mozannar et al., 2023), or perform model routing, selecting among a set of models the one best suited for a given input (Shnitzer et al., 2023). In the LLM setting, several works propose cost-based cascading systems but often lack a principled decision rule to guide deferral (Hu et al., 2024).

# 3 PARTITION-BASED ESTIMATION OF EPISTEMIC UNCERTAINTY

Estimating the grouping loss and epistemic decision risk requires knowledge of the true conditional distribution $f^*(X) = \mathbb{P}[Y = 1 \mid X]$. In practice, however, we only observe discrete outcomes $Y$. Without assumptions on the underlying distributions, a natural approach is to aggregate outcomes to form local average estimates. We adopt this strategy, using a carefully chosen partition to build our estimators. Sections 3.1 and 3.2 assume a fixed partition and introduce estimators for the grouping loss and epistemic decision risk, respectively. Section 3.3 discusses the construction of the partition.

**Notation** Let $\mathcal{L}^{(n)} \stackrel{\text{def}}{=} \{\mathcal{L}_j^{(n)} : j \in [1, J^{(n)}]\}$ be a partition of the input space. For each $j$, let $n_j$ (resp. $\hat{p}_j = \frac{n_j}{n}$) be the number (resp. fraction) of samples belonging to region $j$, and let $p_j \stackrel{\text{def}}{=} \mathbb{P}[X \in \mathcal{L}_j]$

**Definition 3** (Partitioning estimates). *For $(X, R)$ joint random variables in $\mathcal{X} \times \mathbb{R}$, the population level partitioning estimate is defined as:*

$$r_{\mathcal{L}^{(n)}}^*(X) = \sum_{j=1}^{J^{(n)}} r_j^* \mathbf{1}_{X \in \mathcal{L}_j^{(n)}} \quad \text{where} \quad r_j^* = \mathbb{E}[R \mid X \in \mathcal{L}_j^{(n)}] \tag{4}$$

*Let $(X_i, R_i)_{i=1}^n$ be $n$ sample points in $\mathcal{X} \times \mathbb{R}$. The sample level partitioning estimate is defined as:*

$$\hat{r}^{(n)}(X) = \sum_{j=1}^{J^{(n)}} \hat{r}_j^{(n)} \mathbf{1}_{X \in \mathcal{L}_j^{(n)}} \quad \text{where} \quad \hat{r}_j^{(n)} = \frac{1}{n_j} \sum_{X_i \in \mathcal{L}_j^{(n)}} R_i \tag{5}$$

The superscript $(n)$ indicates dependence on a dataset of size $n$. To keep notations lightweight, it will be omitted whenever appropriate.

## 3.1 GROUPING LOSS ESTIMATOR

The grouping loss corresponds to the variance of probabilities $f^*(X)$ within bins of same probability $p$ (eq. (2)). Perez-Lebel et al. (2023)'s grouping loss estimator directly mirrors this definition by binning confidence scores and computing per-bin variances. However, binning comes with several drawbacks. Bin boundaries prevent information from being shared across bins, and level sets corresponding to a bin may form disconnected subsets in feature space $\mathcal{X}$. Both effects may hurt sample efficiency. Deriving binning-free estimators requires departing from the grouping loss expression given in eq. (2), which we achieve in proposition 2.

**Proposition 2** (Grouping loss lower bound, A.1). *Let $\mathcal{L}$ be a partition of the input space and $r_{\mathcal{L}}^*$ the associated population level partitioning estimate applied to $R = Y - c \circ f(X)$. Then the grouping loss is lower-bounded as:*

$$\text{GL} = \mathbb{E}\Big[(f^*(X) - c \circ f(X))^2\Big] \geq \sum_{j=1}^J p_j r_j^{*2} \stackrel{\text{def}}{=} \text{GL}_{lb}(\mathcal{L})$$

$\text{GL}_{lb}(\mathcal{L})$ represents the fraction of the grouping loss that is captured by the partition $\mathcal{L}$. It is defined as the expected squared difference between the outcome and the calibrated score within each region $\mathcal{L}_j$. The tightness of this lower bound depends on how homogeneous the regions are: the smaller the variability of $r(X) = f^*(X) - c \circ f(X)$ within a region, the tighter the lower bound, with equality when $f^* - c \circ f$ is constant over each region. In practice, a good partition should group together points with similar under- or over-confidence. Note that such regions may span multiple confidence-score bins, as similar over/under-confidence levels can occur at different confidence levels $p$.

**Proposition 3** (Debiased grouping loss estimator, A.2). *Given a partition $\mathcal{L}^{(n)}$ and a dataset $\mathcal{D}^{(n)} = \big\{(X_i, Y_i)\big\}_{i=1..n}$, let $\hat{r}_j$ (resp. $\hat{v}_j$) be the sample mean (resp. sample variance) of $Y - c \circ f(X)$ in region $j$. A debiased estimator of $\text{GL}_{lb}(\mathcal{L}^{(n)})$ is:*

$$\widehat{\text{GL}}_{lb}^{(n)} \stackrel{\text{def}}{=} \sum_{j=1}^J \hat{p}_j \left( \hat{r}_j^2 - \frac{1}{n_j} \hat{v}_j \right) \tag{6}$$

The terms involving $\hat{r}_j^2$ are plugin estimators for $\mathrm{GL}_{lb}(\mathcal{L})$, while the terms involving $\hat{v}_j$ provide a bias correction. Importantly, given a partition, the quantities $\hat{r}_j^2$, $\hat{v}_j$ and $\hat{p}_j$ can easily be computed to obtain an estimator of $\mathrm{GL}_{lb}$. While the proposed estimator is a lower bound of the total grouping loss for a given partition, it asymptotically converges to the true total grouping loss GL for a well-suited sequence of partitions:

**Proposition 4** (Consistency of the grouping loss estimator, A.3). *Let $\hat{r}^{(n)}$ be a partitioning estimate applied to $R = Y - c \circ f(X)$. Assume that $\hat{r}^{(n)}$ is weakly universally consistent in $L_2$ towards $r : x \mapsto \mathbb{E}[Y - c \circ f(X) \mid X = x]$, and that the sequence of partitions $\mathcal{L}^{(n)}$ satisfy $J^{(n)}/\sqrt{n} \to 0$. Then the sequence of estimators $\widehat{\mathrm{GL}}_{lb}^{(n)}$ is weakly universally consistent in $L_1$, i.e.,*

$$\mathbb{E}_{\mathcal{D}^{(n)}}\left[\left|\mathrm{GL} - \widehat{\mathrm{GL}}_{lb}^{(n)}\right|\right] \xrightarrow[n \to +\infty]{} 0.$$

Figure 3 experimentally confirms that our estimator convergences to the true value from below as the sample size increases. This consistency result crucially relies on the consistency assumption of $\hat{r}^{(n)}$ towards $r$, which according to Györfi et al. (2002)'s Theorem 4.1 and 4.2 is ensured given two conditions: (i) the maximum diameter of the partition regions vanishes, and (ii) the number of samples per region grows with the total sample size. These assumptions are reasonable for a tree-based partition.

Note that our grouping loss estimator, summed with a calibration error estimator, yields a lower bound of the epistemic loss (Theorem 1).

### 3.2 POINT-WISE EXCESS RISK ESTIMATOR

The epistemic decision risk, defined as $\mathcal{R}_f(x) = EC(\delta_{f,t}, x) - EC(\delta_{f^*,t^*}, x)$ (eq. (3)), quantifies the excess classification cost of a candidate model $f$ relative to the optimal (but unknown) model $f^*$. For pointwise estimation, we focus directly on the total epistemic risk rather than its decomposition into calibration and grouping components. Calibration may improve or degrade individual predictions and only guarantees risk reduction in expectation. Consequently, while binwise or global calibration risks are non-negative, the pointwise calibration risk $\mathcal{R}^{CL}(x)$ isn't always positive. We therefore directly target the pointwise epistemic decision risk and provide below an analytical expression for $\mathcal{R}_f(x)$.

**Proposition 5** (Oracle epistemic decision Risk, A.4). *For any $X \in \mathcal{X}$, the epistemic risk writes:*

$$\mathcal{R}_f(X) = \begin{cases} \Lambda_\Delta \mid f^*(X) - t^* \mid & \text{if } \mathbf{1}_{f^*(X) \geq t^*} \neq \mathbf{1}_{f(X) \geq t} \\ 0 & \text{else.} \end{cases} \tag{7}$$

The epistemic risk is therefore zero whenever $f$ and $f^*$ agree, and otherwise proportional to the distance between $f^*(X)$ and the decision threshold $t^*$. This distance is represented by the red gradient regions in fig. 2. Intuitively, the further away $f^*(X)$ is from the threshold $t^*$, the costlier it is to mispredict. As in grouping loss estimation, $f^*$ is unknown so we approximate the target quantity $\mathcal{R}_f(x)$ via a residual partitioning estimate.

**Proposition 6** (A Partition-Based Pointwise estimation of the Epistemic decision Risk, A.5). *Let $\hat{r}(X)$ be a partitioning estimate (corresponding to partition $\mathcal{L}$) applied to $R = Y - c \circ f(X)$, and let $t^*$ and $\Lambda_\Delta$ as defined in Section 2.3. We define our estimator:*

$$\widehat{\mathcal{R}}_{f,\mathcal{L}}(X) \stackrel{def}{=} \begin{cases} \Lambda_\Delta \mid c \circ f(X) + \hat{r}(X) - t^* \mid & \text{if } \mathbf{1}_{c \circ f(X)) + \hat{r}(X) \geq t^*} \neq \mathbf{1}_{f(X) \geq t} \\ 0 & \text{else.} \end{cases} \tag{8}$$

*Then, for any $X$ belonging to region $j \in [1, J]$:*

$$\left|\mathcal{R}_f(X) - \widehat{\mathcal{R}}_{f,\mathcal{L}}(X)\right| \leq |\Lambda_\Delta|\left|r(X) - \hat{r}_j\right|$$

$\widehat{\mathcal{R}}_{f,\mathcal{L}}(X)$ represents the individual epistemic decision risk of a candidate classifier $f$ captured by the partitioning estimate $\hat{r}$, beyond the calibration risk. Proposition 6 provides a bound on the approximation error, showing that the absolute difference between the oracle and estimated epistemic

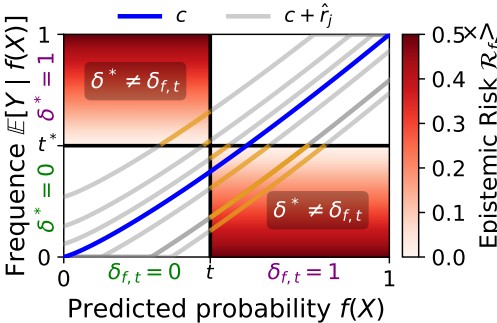

Figure 2: **Epistemic decision risk estimation.** For a model $f$, we plot the calibration curve $c$ (blue) and the region-wise corrections $c + \hat{r}_j$ (grey). The epistemic decision risk is non-zero in the quadrants where the decisions of $f$ and $f^*$ disagree (red quadrants), and grows linearly with the distance to the decision threshold (red gradient). Only points lying in the orange part of the region-wise curves have $\widehat{\mathcal{R}}_{f,\mathcal{L}}(X) > 0$.

decision risk at $X$ is controlled by the distance between the residual $r(X) = f^*(X) - c \circ f(X)$ and its partitioning estimate $\hat{r}_j$. In practice, tree partitions are constructed to minimize the intra-leaf variance of the residuals, thereby tightening this control. Figure 2 illustrates our epistemic risk estimator: each region $\hat{r}_j$ provides a localized correction to calibrated scores, enabling a refined mapping of individual points to their corresponding decision epistemic risk estimates.

The pointwise nature of $\widehat{\mathcal{R}}_{f,\mathcal{L}}(X)$ enables estimating the potential accuracy loss for each individual input, rather than on average over level sets or the entire data distribution. This property is crucial for decision making, as in our LLM cascading experiments (section 4.3), with a per-sample risk estimation to guide model selection. As for the grouping loss, this estimator converges to the true value of the epistemic risk if it is computed from a sequence of consistent partitioning estimates.

**Proposition 7** (Consistency of the epistemic decision risk estimator, A.6). *Let $\hat{r}^{(n)}$ be a partitioning estimate applied to $R = Y - c \circ f(X)$. Assume that $\hat{r}^{(n)}$ is weakly universally consistent in $L_2$. Then the sequence of estimators $\widehat{\mathcal{R}}_{f,\mathcal{L}^{(n)}}^{(n)}$ is weakly universally consistent in $L_2$, i.e.,*

$$\mathbb{E}_{\mathcal{D}^{(n)}} \left[ \mathbb{E}_X \left[ \, | \, \widehat{\mathcal{R}}_{f,\mathcal{L}^{(n)}}^{(n)}(X) - \mathcal{R}_f(X) \, |^2 \, \right] \right] \xrightarrow[n \to +\infty]{} 0.$$

### 3.3 Estimators in Practice

**Partitioning estimate** - Our estimators $\widehat{\mathrm{GL}}_{lb}$ and $\widehat{\mathcal{R}}_{f,\mathcal{L}}(X)$ both rely on a partitioning estimate $\hat{r}$. To obtain $\hat{r}$ we use *honest* regression trees (Wager & Athey, 2017). They mitigate bias in leaf estimates by ensuring that each sample is used either to fit the tree structure or to compute the leaf-level estimates, but never both. Concretely, we fit a regression tree on the model residuals after calibration, i.e., $Y - c \circ f(X)$, using one subset of the data. The resulting leaves define the partition. The estimates within each leaf are then computed on a disjoint subset. Algorithm 1 details this procedure. Consistent with Proposition 6, the tree partitions are built to minimize the intra-leaf variance of the residuals. Under some assumptions (Appendix A.7), honest trees are a consistent partitioning estimate. Finally, while boosting models typically outperform single trees in predictive accuracy, simple honest trees provide a lower-variance and more interpretable partitioning, which is preferable for evaluation purposes under limited sample sizes. Results with more sophisticated residual models are presented in Appendix C.8.

**Implementation aspects** - For calibration, we use Platt scaling (Platt, 1999). The partitioning estimate uses the same hyperparameters across all experiments to ensure robustness: decision trees with unlimited depth and a minimum of 15 samples per leaf at training time. Appendix C.6 experimentally investigates the influence of tree depth. Datasets are divided into a calibration set (10%), a fitting set (40%), and an evaluation set (50%) to estimate GL or fit the epistemic decision risk estimator. The choice of input space for the tree is determined by the data type. For standard tabular data, the original feature space is used. In the case of neural networks handling unstructured data like images or text, the partition is constructed from the model's internal representations, such as the activations from the last hidden layer (Perez-Lebel et al., 2023) or a learned low-dimensional UMAP embedding (Detommaso et al., 2024). Appendix B.3 further details our proposed implementation.

## 4 EXPERIMENTS

### 4.1 VALIDATION OF THE GROUPING LOSS ESTIMATOR ON SEMI-SYNTHETIC DATA

We evaluate the grouping loss estimator (eq. (6)) on a semi-synthetic dataset with ground-truth knowledge of $f^*(X)$. We use the Weather dataset from TabReD (Rubachev et al., 2025), which contains more than 16 million samples and exhibits temporal drift—a favorable setting to study grouping loss and consistency. Appendix B.4 details data construction.

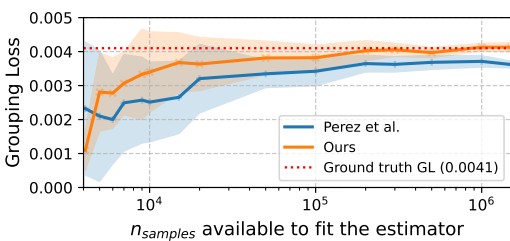

Figure 3: **Sample efficiency and asymptotic consistency of** $\widehat{\mathrm{GL}}_{lb}^{(n)}$ on a semi-synthetic experiment.

Confidence scores $f$ are obtained from a Histogram Gradient Boosted tree trained on the training split. These scores are then evaluated using our estimator $\widehat{\mathrm{GL}}_{lb}$ as well as the method proposed in Perez-Lebel et al. (2023). Figure 3 confirms that $\widehat{\mathrm{GL}}_{lb}$ indeed provides a lower bound on the ground-truth grouping loss in expectation. Furthermore, we observe its asymptotic consistency: as the number of samples increases, the estimator converges to the true grouping loss. Finally, our estimator demonstrates greater sample efficiency compared to Perez-Lebel et al. (2023), capturing more grouping loss for a given sample size. This highlights that a binning-free estimator can leverage available data more effectively to capture the remaining heterogeneity in confidence scores.

**Quantifying suboptimality** The excess decision risk introduced in section 2.3 provides a principled way to quantify overall suboptimality. Following the experimental setup of Perez-Lebel et al. (2025), we show in Appendix C.1 that our decision risk estimator correlates more strongly with the average cost reductions achieved through various fine-tuning strategies than theirs.

### 4.2 LLMs' GROUPING LOSS: OVER- AND UNDER-CONFIDENCE ACROSS GROUPS

To evaluate LLM confidence scores, we use Folktexts (Cruz et al., 2024), a benchmark designed to evaluate large language models (LLMs) on question-answering tasks with aleatoric uncertainty. Folktexts is constructed by converting tabular data from Folktables (Ding et al., 2021) into natural language questions with binary-choice answers. Each prompt includes general context, a description of an individual row (formatted as "The {column name} is {value}" for each column), and the question of interest. We use all available tasks from Folktexts (ACSIncome, ACSEmployment, ACSMobility, ACSTravelTime, and ACSPublicCoverage), each involving a binary choice setting based on socio-economic information. These tasks inherently involve aleatoric uncertainty, as the outcome cannot be determined with certainty from the available information. We use 10% of the data for all datasets except ACSEmployment, for which we use 5%. Confidence scores are derived from next-token probabilities of candidate answers. If the possible outcomes are yes and no, the confidence score is defined as $\frac{\mathbb{P}(\text{yes})}{\mathbb{P}(\text{yes})+\mathbb{P}(\text{no})}$.

We evaluate the *zero-shot* confidence scores of 27 open-source LLMs (listed in Appendix B.5), whose grouping loss are summarized in fig. 4. Open-source models are required to access next-token probabilities used to compute confidence scores. The selection covers model sizes from 1B to 70B parameters, sampled as uniformly as possible in this range, and includes families with multiple scales such as Llama 3 (1B, 3B, 8B, 70B). Models are evaluated using our estimator $\widehat{\mathrm{GL}}_{lb}$.

Figure 4 reveals a clear size effect: grouping loss decreases with model size, suggesting that larger models provide more reliable confidence estimates. Another consistent trend emerges: across all sizes, instruction-tuned models exhibit lower (or equal) grouping loss compared to their base counterparts, indicating that instruction-tuning may reduce subgroup-specific biases. This observation complements the calibration analysis of Cruz et al. (2024), which shows that instruction-tuning polarizes answer distributions and induces over-confidence. Appendix C.2 provides the non-normalized grouping loss estimates per dataset, highlighting that LLMs display substantial confidence biases regardless of model size. A closer inspection of the learned groups for Llama 3.3 70B Instruction-tuned (fig. 1) reveals systematic biases for certain demographic subgroups, such as underestimating the probability

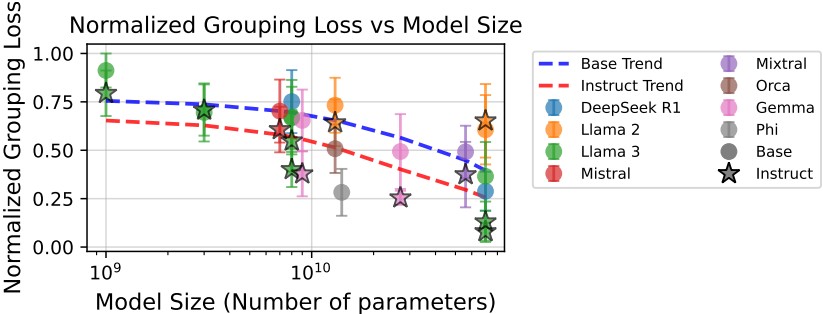

Figure 4: **Normalized Grouping loss as a function of LLM size, for base and instruct models.** For each (LLM, dataset) pair spanning the 27 LLMs and 5 ACS datasets, GL estimates were averaged over 5 random seeds governing the calibration, fitting, and evaluation splits. The resuting GL estimation was min-max normalized across models for each dataset, and we report the mean and standard error per model, across datasets. The base and instruct trends were computed using a LOWESS regression.

of high income for experienced graduate women, even after calibration. These results echo the findings of Cruz et al. (2024) highlighting biases for Black and White subpopulations.

## 4.3 ESTIMATING EPISTEMIC RISK IMPROVES LLM CASCADES

A key strength of our risk estimator is that it captures the quality of individual confidence scores. We illustrate this by using our estimator for LLM routing, i.e. selecting from a pool of expert models the one expected to perform best at the lowest cost (Hu et al., 2024).

**Baseline: Predictive router** Predictive routing selects an LLM from a pool *without first generating outputs* for the incoming query (Shnitzer et al., 2023). Formally, the accuracy of model $m$ on query $x$, denoted $Acc_m(x)$, is estimated by training on past queries labeled by whether the model's answer was correct. Hu et al. (2024) implements this with kNN and MLP routers trained on query embeddings. Since our estimators rely on decision trees applied to structured query features, we trained a decision-tree router in the same feature space, using identical hyperparameters and the same total number of training samples as the risk estimator (see section 3.3) to ensure a fair comparison (see Appendix C.4 for complementary results using query embeddings). At inference time, given a new query $x$, the router selects the model that best balances expected accuracy and cost: $\arg\max_{m \in \text{LLM pool}} \lambda Acc_m(x) - \text{cost}_m$.

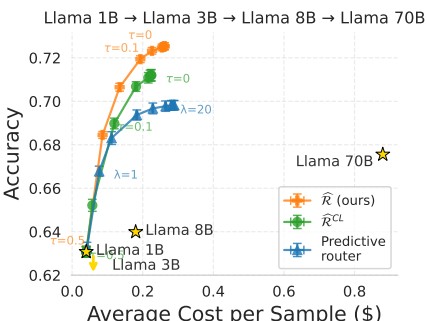

Figure 5: **Cost-accuracy tradeoffs** on the ACSIncome dataset when varying the threshold of risk $\tau$ and willingness to pay $\lambda$ for the Llama 3 instruction-tuned cascade.

Here, $\lambda$ encodes the user's *willingness to pay*: larger values favor more accurate, potentially more expensive models.

**Cascades:** Cascades provide an alternative routing strategy to predictive routing. A conventional confidence-based cascade sequentially queries a list of models (here, LLMs). At each step, the current model is queried to get a confidence score: if it exceeds a user-defined threshold $\tau$ for one class, the model is selected; otherwise, the next model is evaluated (Jitkrittum et al., 2023). Compared to predictive routing, costs accumulate each time a new model is queried since a forward pass is required to get a confidence score, whereas predictive routing queries only a single model per input. However, this strategy proved ineffective in our experiments, as LLMs are poorly calibrated (see Appendix C.5)

**Epistemic decision risk cascade (ours):** Instead of directly relying on confidence scores thresholding, we employ our epistemic risk estimate, $\widehat{\mathcal{R}}_{LLM_m}(x)$, to rule for deferral. This replaces the oracle quality scores used by Hu et al. (2024)—addressing the difficulty of designing effective scoring functions—and serves as the metric for the cascade process. Appendix B.6 illustrates our proposal.

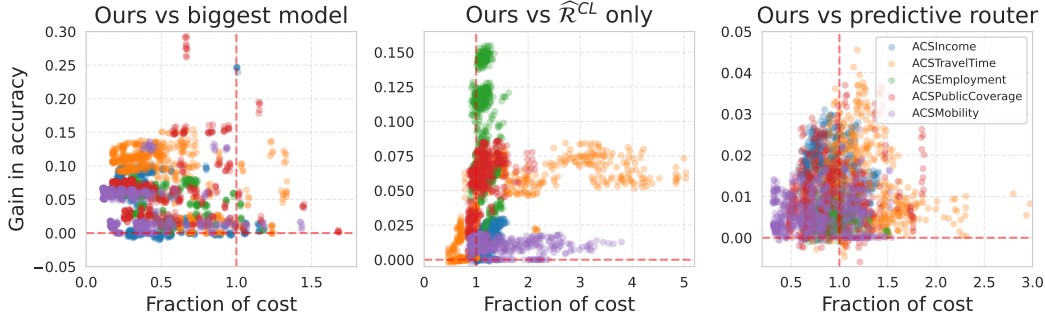

Figure 6: **Cascades: our risk estimator gives better accuracies at a lower cost.** Each point corresponds to a cascade (an LLM sequence). The y-axis is Acc(ours) - Acc(baseline). Values are positive when our cascade improves accuracy over the baseline. The x-axis is Cost(ours)/Cost(baseline). For values below 1, our model is less expensive. Our cascade based on $\widehat{\mathcal{R}}$ ($\tau = 0$) is compared to **Left:** the biggest LLM in the cascade, **Middle:** A cascade based on $\widehat{\mathcal{R}}^{CL}$ only ($\tau = 0$). **Right:** the predictive router router baseline ($\lambda = 100$).

If no model satisfies the threshold, the model with the lowest risk is chosen. We compare cascades built using our epistemic risk estimator $\widehat{\mathcal{R}}$ to a baseline cascade that relies on the bin-wise calibration risk $\mathcal{R}^{CL}$ (Appendix B.1) for individual deferral decisions. This baseline highlights the difference between our approach and a strategy based solely on calibration to guide deferral.

Figure 5 shows the cascades and predictive router performances for the LLama 3 instruction-tuned pool of models, on the ACSIncome dataset. For high willingness to pay $\lambda$ (equivalently, low threshold $\tau$), our cascade outperforms the largest model in the pool – +4% in accuracy compared to Llama 70B – at about 30% of its cost. This indicates that smaller models were appropriately retained for some inputs. Moreover, our cascade based on $\widehat{\mathcal{R}}_{LLM_m}(x)$ (shortened as $\widehat{\mathcal{R}}$ in the legend) systematically boosts accuracy by up to 2% compared to the two competing approaches at any given cost.

Figure 6 summarizes performances across the 5 Folktexts datasets and 120 different cascade pools built from the Llama 3 instruction tuned family (1B, 3B, 8B, 70B), Phi4 (14B), Gemma 2 (27B) and Mixtral8x7B (ordered by size). Performances are reported for $\tau = 0$. As shown in fig. 6 (left), our cascade consistently improves upon the accuracy of the largest model at a fraction of its cost (on average, a 6% accuracy gain at 46% of the cost). Figure 6 (middle) shows that while the calibration risk $\mathcal{R}_f^{CL}$ provides a useful score (as seen in fig. 5), taking into account the full epistemic risk allows for high accuracy gains. This is particularly pronounced when models exhibit high grouping loss, such as on the ACSEmployment dataset (see Appendix C.2 for grouping loss values) where accounting for the full epistemic risk leads to accuracy gains of up to 15%. Finally, fig. 6 (right) shows that our cascades systematically outperform the predictive router baseline and achieve strict Pareto optimality (higher accuracy at lower cost, i.e. left upper quadrant) in 60% of cases.

**Effect of the number of samples**     These results are robust to training set size (Appendix C.7). With more than 6,000 samples to train the risk estimator (either $\widehat{\mathcal{R}}$ or $\widehat{\mathcal{R}}^{CL}$), accounting for full epistemic risk consistently improves cascade accuracy or cost over calibration-only risk across all datasets. Below 6,000 samples, performance gains depend on the grouping loss magnitude. For high-grouping-loss datasets (e.g., ACSEmployment), full epistemic risk remains beneficial even in low-data regimes, yielding up to +12% accuracy with only 2,000 samples.

## 5 CONCLUSION: BETTER EPISTEMIC EVALUATION GIVES BETTER DECISIONS

Additional data can be used either for fine-tuning or for evaluation, analogous to recalibration versus estimating calibration error. While the trade-offs depend on the downstream tasks, evaluation is indispensable. We introduced estimators targeting the total epistemic loss and associated excess decision risk. They improve sample efficiency while revealing model biases thanks to an interpretable learned partition. Importantly, the excess risk estimator operates at the sample level, enabling selective deferral of specific cases to a stronger model or a human at a controlled cost. This property is particularly valuable when deploying LLMs in complex or high-risk environments.

## ACKNOWLEDGMENTS

This work received funding from Inria's *Action Exploratoire* AuditAI and was performed using HPC resources from GENCI–IDRIS (Grant 2025-AD011016418). The authors thank Tristan Haugomat for helping refine the final manuscript.

## REPRODUCIBILITY STATEMENT

The source code is available on GitHub[1]. All datasets and open-source models are publicly available with references in Appendix B. For the semi-synthetic experiment of section 4.1, dataset modifications details are given in Appendix B.4. Finally, the proofs of all theoretical results are listed in Appendix A.

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

# Appendix

## Table of Contents

# A  PROOFS

## A.1  PROOF OF PROPOSITION 2: GROUPING LOSS LOWER-BOUND

**Proposition 2** (Grouping loss lower bound, A.1). *Let $\mathcal{L}$ be a partition of the input space and $r^*_{\mathcal{L}}$ the associated population level partitioning estimate applied to $R = Y - c \circ f(X)$. Then the grouping loss is lower-bounded as:*

$$\text{GL} = \mathbb{E}\Big[(f^*(X) - c \circ f(X))^2\Big] \geq \sum_{j=1}^{J} p_j r^{*2}_j \overset{def}{=} \text{GL}_{lb}(\mathcal{L})$$

*Proof.* We recall that:

$$r^*_j = \mathbb{E}[Y - c \circ f(X) \mid X \in \mathcal{L}^{(n)}_j]$$

and $p_j = \mathbb{P}[X \in \mathcal{L}_j]$.

We have:

$$\text{GL} = \mathbb{E}_X\left[(f^*(X) - c \circ f(X))^2\right] \tag{9}$$

$$= \mathbb{E}_j[\mathbb{E}[(f^*(X) - c \circ f(X))^2 \mid X \in \mathcal{L}^{(n)}_j]] \tag{10}$$

$$\geq \mathbb{E}_j[\mathbb{E}[f^*(X) - c \circ f(X) \mid X \in \mathcal{L}^{(n)}_j]^2] \tag{11}$$

$$\geq \mathbb{E}_j[\mathbb{E}[Y - c \circ f(X) \mid X \in \mathcal{L}^{(n)}_j]^2] \tag{12}$$

$$\geq \sum_{j=1}^{J} p_j r^{*2}_j \tag{13}$$

(10) Total Expectation ($\mathcal{L}$ is a partition of $\mathcal{X}$)

(11) Jensen's inequality to the function $x \mapsto x^2$

(12) $Y$ is binary and $Y \sim \mathcal{B}(f^*(X))$

(13) By definition of $r^*_j$ and $p_j$

Which concludes the proof. □

## A.2  PROOF OF PROPOSITION 3: DEBIASING THE GROUPING LOSS ESTIMATOR

**Proposition 3** (Debiased grouping loss estimator, A.2). *Given a partition $\mathcal{L}^{(n)}$ and a dataset $\mathcal{D}^{(n)} = \left\{(X_i, Y_i)\right\}_{i=1..n}$, let $\hat{r}_j$ (resp. $\hat{v}_j$) be the sample mean (resp. sample variance) of $Y - c \circ f(X)$ in region $j$. A debiased estimator of $\text{GL}_{lb}(\mathcal{L}^{(n)})$ is:*

$$\widehat{\text{GL}}^{(n)}_{lb} \overset{def}{=} \sum_{j=1}^{J} \hat{p}_j \left(\hat{r}^2_j - \frac{1}{n_j}\hat{v}_j\right) \tag{6}$$

The proof is inspired from the proof of proposition 4.3 in Perez-Lebel et al. (2023).

*Proof.* We note

$$\widehat{\text{GL}}_{plugin} \overset{def}{=} \sum_{j=1}^{J} \hat{p}_j \hat{r}^2_j$$

the plugin version of the estimator from the definition of $\text{GL}_{lb}$ in Proposition 2.

We have

$$\mathbb{E}\left[\widehat{\text{GL}}_{plugin}\right] = \sum_{j=1}^{J} \mathbb{E}[\hat{p}_j \hat{r}^2_j]$$

In the case where region $j$ is empty (i.e. $n_j = 0$), $\hat{p}_j = 0$ and $\hat{r}_j$ is undefined. We therefore set $\hat{p}_j \hat{r}_j = 0$ in this case. Hence, for $j \in [1, J]$ fixed, we have:

$$
\begin{aligned}
\mathbb{E}[\hat{p}_j \hat{r}_j^2] &= \mathbb{E}\left[\mathbb{E}[\hat{p}_j \hat{r}_j^2 \mathbf{1}_{\hat{p}_j \geq 0} \mid \hat{p}_j]\right] \quad \text{total expectation} \\
&= \mathbb{E}\left[\hat{p}_j \mathbf{1}_{\hat{p}_j \geq 0} \mathbb{E}[\hat{r}_j^2 \mid \hat{p}_j]\right]
\end{aligned}
$$

with:

$$
\mathbb{E}[\hat{r}_j^2 \mid \hat{p}_j] = \mathbb{E}[\hat{r}_j \mid \hat{p}_j]^2 + \mathbb{V}[\hat{r}_j \mid \hat{p}_j]
$$

Furthermore

$$
\mathbb{E}[\hat{r}_j \mid \hat{p}_j] = \mathbb{E}\left[\frac{1}{n_j} \sum_{X_i \in \mathcal{L}_j} Y_i - c \circ f(X_i) \mid \hat{p}_j\right] \tag{14}
$$

$$
= \frac{1}{n_j} \mathbb{E}\left[\sum_{X_i \in \mathcal{L}_j} Y_i - c \circ f(X_i)\right] \tag{15}
$$

$$
= \frac{1}{n_j} \sum_{i=1}^{n_j} \mathbb{E}[Y_i - c \circ f(X_i) \mid X_i \in \mathcal{L}_j] \tag{16}
$$

$$
= \mathbb{E}[Y - c \circ f(X) \mid X \in \mathcal{L}_j] \tag{17}
$$

(14): by definition.

(16): linearity of expectation.

(17): $Y_i - c \circ f(X_i)$ are identically distributed $\mid \mathcal{L}_j$.

And

$$
\mathbb{V}[\hat{r}_j \mid \hat{p}_j] = \mathbb{V}\left[\frac{1}{n_j} \sum_{X_i \in L_j} Y_i - c \circ f(X_i) \mid \hat{p}_j\right] \tag{18}
$$

$$
= \frac{1}{n_j^2} \mathbb{V}\left[\sum_{X_i \in L_j} Y_i - c \circ f(X_i)\right] \tag{19}
$$

$$
= \frac{1}{n_j^2} \sum_{i=1}^{n_j} \mathbb{V}[Y_i - c \circ f(X_i) \mid X_i \in \mathcal{L}_j] \tag{20}
$$

$$
= \frac{1}{n_j} \mathbb{V}[Y - c \circ f(X) \mid X \in \mathcal{L}_j] \tag{21}
$$

(18): by definition.

(20): $(Y_i - c \circ f(X_i))$ are independent.

(21): $Y_i - c \circ f(X_i)$ are identically distributed $\mid \mathcal{L}_j$.

Which gives:

$$\mathbb{E}[\hat{p}_j \hat{r}_j^2] = \mathbb{E}\left[\hat{p}_j \mathbf{1}_{\hat{p}_j \geq 0} \mathbb{E}[\hat{r}_j^2 \mid \hat{p}_j]\right]$$

$$= \mathbb{E}\left[\hat{p}_j \mathbf{1}_{\hat{p}_j \geq 0}\left(\mathbb{E}[Y - c \circ f(X) \mid X \in \mathcal{L}_j]^2 + \frac{1}{n_j}\mathbb{V}[Y - c \circ f(X) \mid X \in \mathcal{L}_j]\right)\right]$$

$$= p_j \mathbb{E}[Y - c \circ f(X) \mid X \in \mathcal{L}_j]^2 + \mathbb{E}\left[\frac{1}{n}\mathbb{V}[Y - c \circ f(X) \mid X \in \mathcal{L}_j]\mathbf{1}_{\hat{p}_j \geq 0}\right]$$

$$= p_j \mathbb{E}[Y - c \circ f(X) \mid X \in \mathcal{L}_j]^2 + \frac{1}{n}(1 - \nu_j)\mathbb{V}[Y - c \circ f(X) \mid X \in \mathcal{L}_j]$$

$$= p_j \left(\underbrace{\mathbb{E}[Y - c \circ f(X) \mid X \in \mathcal{L}_j]^2}_{\text{plugin term}} + \underbrace{\frac{1}{n_j}(1 - \nu_j)\mathbb{V}[Y - c \circ f(X) \mid X \in \mathcal{L}_j]}_{\text{bias term}}\right)$$

With $\nu_j \overset{\text{def}}{=} (1 - p_j)^n$, the probability that no samples falls in region $\mathcal{L}_j$. We make the approximation that $\nu_j \approx 0$, which is correct when $n$ and $p_j$ are sufficiently large. Therefore, we debias the plugin estimate leading to our estimator:

$$\widehat{\text{GL}}_{lb}(\mathcal{L}) = \sum_{j=1}^{J} \hat{p}_j \left(\hat{r}_j^2 - \frac{1}{n_j}\hat{v}_j\right)$$

with the unbiased sample variance (i.e. with Bessel correction) estimate:

$$\hat{v}_j \overset{\text{def}}{=} \frac{1}{n_j - 1}\sum_{i \in \mathcal{L}_j}\left(Y_i - c \circ f(X_i) - \frac{1}{n_j}\sum_{i \in \mathcal{L}_j}Y_i - c \circ f(X_i)\right)^2$$

$\square$

### A.3 PROOF OF PROPOSITION 4: CONSISTENCY OF THE GROUPING LOSS ESTIMATOR

Recall that $r$ is defined as $r : x \mapsto \mathbb{E}[Y - c \circ f(X) \mid X = x]$, while $\hat{r}(X)$ is a region-wise constant approximation, as in defined in eq. (5).

Recall that by definition:

$$\text{GL} = \mathbb{E}_X[(f^*(X) - c \circ f(X))^2]$$

$$= \mathbb{E}_X\left[\mathbb{E}[Y - c \circ f(X) \mid X]^2\right]$$

$$= \mathbb{E}_X[r(X)^2]$$

And we defined:

$$\widehat{\text{GL}}_{lb}^{(n)} \overset{\text{def}}{=} \sum_{j=1}^{J} \hat{p}_j \left(\hat{r}_j^2 - \frac{1}{n_j}\hat{v}_j\right)$$

in eq. (6).

We introduce the true expectation of the sample level partitioning estimate as an intermediate value:

$$\text{GL}_{lb}(\mathcal{L}^{(n)}, \mathcal{D}^{(n)}) \overset{\text{def}}{=} \mathbb{E}_X[\hat{r}^2(X)] \tag{22}$$

$$= \mathbb{E}_j[\hat{r}_j^2] \tag{23}$$

$$= \sum_j p_j \hat{r}_j^2 \tag{24}$$

With $p_j = \mathbb{P}[X \in \mathcal{L}_j^{(n)}]$. For the proof of Proposition 4, we require two lemmas: lemma 1 showing the consistency of $\text{GL}_{lb}(\mathcal{L}^{(n)}, \mathcal{D}^{(n)})$ towards GL, and lemma 2 for the consistency of $\widehat{\text{GL}}_{lb}^{(n)}$ towards $\text{GL}_{lb}(\mathcal{L}^{(n)}, \mathcal{D}^{(n)})$.

**Lemma 1.** *Let $(\mathcal{L}^{(n)})_{n\in\mathbb{N}}$ be a sequence of partitions such that the associated partitioning estimate $\hat{r}^{(n)}$ is weakly universally consistent in $L_2$ towards $r$* [2]*. Then $\mathrm{GL}_{lb}(\mathcal{L}^{(n)}, \mathcal{D}^{(n)})$ is weakly universally consistent in $L_1$ towards* GL*:*

$$\mathbb{E}_{\mathcal{D}^{(n)}}\left[\mid \mathrm{GL} - \mathrm{GL}_{lb}(\mathcal{L}^{(n)}, D^{(n)}) \mid\right] \xrightarrow[n\to+\infty]{} 0$$

*Proof.* The partitioning estimate $\hat{r}^{(n)}$ is consistent towards

$$r : x \mapsto \mathbb{E}[Y - c \circ f(X) \mid X = x],$$

that is, from Györfi et al. (2002)'s definition 1.1. page 13:

$$\mathbb{E}_{\mathcal{D}^{(n)}}\left[\mathbb{E}_X\left[(\hat{r}^{(n)}(X) - r(X))^2\right]\right] \xrightarrow[n\to+\infty]{} 0$$

and from (24):

$$\mathrm{GL}_{lb}(\mathcal{L}^{(n)}, \mathcal{D}^{(n)}) = \mathbb{E}_X[\hat{r}^{(n)}(X)^2]$$

We compute:

$$\mid \mathrm{GL} - \mathrm{GL}_{lb}(\mathcal{L}^{(n)}, \mathcal{D}^{(n)}) \mid^2 = \mid \mathbb{E}_X[r(X)^2 - \hat{r}^{(n)}(X)^2] \mid^2 \tag{25}$$

$$= \mid \mathbb{E}_X[(r(X) - \hat{r}^{(n)}(X))(r(X) + \hat{r}^{(n)}(X))] \mid^2 \tag{26}$$

$$\leq \mathbb{E}_X[\mid r(X) - \hat{r}^{(n)}(X) \mid^2]\mathbb{E}_X[\mid r(X) + \hat{r}^{(n)}(X) \mid^2] \tag{27}$$

$$\leq 4\mathbb{E}_X[\mid r(X) - \hat{r}^{(n)}(X) \mid^2] \tag{28}$$

(27): Cauchy Schwarz.

(28): $\mid r(X) + \hat{r}^{(n)}(X) \mid \leq 2$ by definition.

So taking the expectation over $\mathcal{D}^{(n)}$, we have:

$$\mathbb{E}_{\mathcal{D}^{(n)}}\left[\mid \mathrm{GL} - \mathrm{GL}_{lb}(\mathcal{L}^{(n)}, \mathcal{D}^{(n)}) \mid^2\right] \leq 4\mathbb{E}_{\mathcal{D}^{(n)}}\left[\mathbb{E}_X[\mid r(X) - \hat{r}^n(X) \mid^2]\right] \xrightarrow[n\to+\infty]{} 0$$

by consistency hypothesis. Finally, we use Cauchy Schwarz to obtain a result in $L_1$ norm:

$$\mathbb{E}_{\mathcal{D}^{(n)}}\left[\mid \mathrm{GL} - \mathrm{GL}_{lb}(\mathcal{L}^{(n)}, \mathcal{D}^{(n)}) \mid\right] \leq \sqrt{\mathbb{E}_{\mathcal{D}^{(n)}}\left[\mid \mathrm{GL} - \mathrm{GL}_{lb}(\mathcal{L}^{(n)}, \mathcal{D}^{(n)}) \mid^2\right]} \xrightarrow[n\to+\infty]{} 0$$

$\square$

**Lemma 2.** *Let $(\mathcal{L}^{(n)})_{n\in\mathbb{N}}$ be a sequence of partitions such that the associated partitioning estimate $\hat{r}^{(n)}$ is weakly universally consistent in $L_2$ towards $r$ with a sequence of partitions $\mathcal{L}^{(n)}$ s.t. $J^{(n)}/\sqrt{n} \to 0$. $\widehat{\mathrm{GL}}_{lb}(\mathcal{L}^{(n)}, \mathcal{D}^{(n)})$ is weakly universally consistent towards $\mathrm{GL}_{lb}(\mathcal{L}^{(n)}, \mathcal{D}^{(n)})$ in $L_1$:*

$$\mathbb{E}_{\mathcal{D}^{(n)}}\left[\mid \mathrm{GL}_{lb}(\mathcal{L}^{(n)}, \mathcal{D}^{(n)}) - \widehat{\mathrm{GL}}_{lb}(\mathcal{L}^{(n)}) \mid\right] \xrightarrow[n\to+\infty]{} 0$$

---

[2] weakly universally consistent according to Györfi et al. (2002)'s definition 1.1. page 13

*Proof.*

$$| \widehat{\mathrm{GL}}_{lb}(\mathcal{L}^{(n)}) - \mathrm{GL}_{lb}(\mathcal{L}^{(n)}, \mathcal{D}^{(n)}) | = \left| \sum_{j=1}^{J_n} \hat{p}_j (\hat{r}_j^{(n)})^2 - \frac{1}{n} \hat{v}_j^{(n)} - \sum_{j=1}^{J_n} p_j (\hat{r}_j^{(n)})^2 \right| \tag{29}$$

$$\leq \left| \sum_{j=1}^{J_n} \hat{p}_j (\hat{r}_j^{(n)})^2 - p_j (\hat{r}_j^{(n)})^2 \right| + \left| \sum_{j=1}^{J_n} \frac{1}{n} \hat{v}_j^{(n)} \right| \tag{30}$$

$$\leq \sum_{j=1}^{J_n} | \hat{p}_j - p_j | (\hat{r}_j^{(n)})^2 + \left| \sum_{j=1}^{J_n} \frac{1}{n} \hat{v}_j^{(n)} \right| \tag{31}$$

$$\leq \sum_{j=1}^{J_n} | \hat{p}_j - p_j | (\hat{r}_j^{(n)})^2 + \frac{J_n}{n} \tag{32}$$

(29): By definition of both terms.

(30): Triangle inequality.

(31): Triangle inequality.

(32): $| Y - c \circ f(X) | \leq 1$ so $\hat{v}_j^{(n)} \leq 1$.

Taking the expectation over $\mathcal{D}^{(n)}$:

$$\mathbb{E}_{\mathcal{D}^{(n)}}[| \widehat{\mathrm{GL}}_{lb}(\mathcal{L}^{(n)}) - \mathrm{GL}_{lb}(\mathcal{L}^{(n)}, \mathcal{D}^{(n)}) |] \leq \mathbb{E}_{\mathcal{D}^{(n)}} \left[ \sum_{j=1}^{J_n} | \hat{p}_j - p_j | (\hat{r}_j^{(n)})^2 + \frac{J_n}{n} \right] \tag{33}$$

$$\leq \sum_{j=1}^{J_n} \mathbb{E}_{\mathcal{D}^{(n)}}[| \hat{p}_j - p_j | (\hat{r}_j^{(n)})^2] + \frac{J_n}{n} \tag{34}$$

$$\leq \sum_{j=1}^{J_n} \mathbb{E}_{\mathcal{D}^{(n)}}[| \hat{p}_j - p_j |] + \frac{J_n}{n} \tag{35}$$

$$\leq \sum_{j=1}^{J_n} \sqrt{\mathbb{E}_{\mathcal{D}^{(n)}}[| \hat{p}_j - p_j |^2]} + \frac{J_n}{n} \tag{36}$$

$$\leq \sum_{j=1}^{J_n} \sqrt{\mathbb{V}_{\mathcal{D}^{(n)}}[\hat{p}_j]} + \frac{J_n}{n} \tag{37}$$

$$\leq \sum_{j=1}^{J_n} \sqrt{\frac{p_j(1 - p_j)}{n}} + \frac{J_n}{n} \tag{38}$$

$$\leq \frac{J_n}{2\sqrt{n}} + \frac{J_n}{n} \tag{39}$$

$$\leq \frac{3 J_n}{2\sqrt{n}} \xrightarrow[n \to +\infty]{} 0 \tag{40}$$

(34): $\frac{J_n}{n}$ doesn't depend on $\mathcal{D}^{(n)}$.

(35): $| Y - c \circ f(X) | \leq 1$ so $\hat{r}_j^{(n)} \leq 1$.

(36): Cauchy Schwarz.

(37): $\mathbb{E}_{\mathcal{D}^{(n)}}[\hat{p}_j] = p_j$.

(38): Variance of a Bernoulli randome variable.

(39): $p_j(1 - p_j) \leq \frac{1}{4}$ for $p_j \in [0, 1]$.

(40): $J^{(n)}/\sqrt{n} \to 0$. $\qquad\qquad\qquad\qquad\qquad\qquad\qquad\qquad\qquad\qquad\qquad\qquad$ □

We now prove the final result.

**Proposition 4** (Consistency of the grouping loss estimator, A.3). *Let $\hat{r}^{(n)}$ be a partitioning estimate applied to $R = Y - c \circ f(X)$. Assume that $\hat{r}^{(n)}$ is weakly universally consistent in $L_2$ towards $r : x \mapsto \mathbb{E}[Y - c \circ f(X) \mid X = x]$, and that the sequence of partitions $\mathcal{L}^{(n)}$ satisfy $J^{(n)}/\sqrt{n} \to 0$. Then the sequence of estimators $\widehat{\mathrm{GL}}_{lb}^{(n)}$ is weakly universally consistent in $L_1$, i.e.,*

$$\mathbb{E}_{\mathcal{D}^{(n)}} \left[ \left| \mathrm{GL} - \widehat{\mathrm{GL}}_{lb}^{(n)} \right| \right] \xrightarrow[n \to +\infty]{} 0.$$

*Proof.* Combining Lemma 1 and Lemma 2, and using the triangle inequality, we have the result.

$\qquad\qquad\qquad\qquad\qquad\qquad\qquad\qquad\qquad\qquad\qquad\qquad\qquad\qquad\qquad\qquad\qquad$ □

### A.4 PROOF OF PROPOSITION 5: ORACLE EPISTEMIC RISK EXPRESSION.

We use Perez-Lebel et al. (2025)'s corollary B.4 recalled below to derive an expression of $\mathcal{R}_f(X)$:

**Corollary** (Difference in expected utilities, (Perez-Lebel et al., 2025)). *Let $\delta_1, \delta_2 \in \mathcal{X} \to \{0, 1\}$ and $x \in \mathcal{X}$. Then the difference in expected costs (defined as the opposite of the expected utilities in Perez-Lebel et al. (2025)) of $\delta_1$ and $\delta_2$ conditional to $x$ is:*

$$EC(\delta_2, x) - EC(\delta_1, x) = \Lambda_\Delta(\delta_1(x) - \delta_2(x))(f^*(x) - t^*)$$

**Proposition 5** (Oracle epistemic decision Risk, A.4). *For any $X \in \mathcal{X}$, the epistemic risk writes:*

$$\mathcal{R}_f(X) = \begin{cases} \Lambda_\Delta \mid f^*(X) - t^* \mid & \text{if } \mathbf{1}_{f^*(X) \geq t^*} \neq \mathbf{1}_{f(X) \geq t} \\ 0 & \text{else.} \end{cases} \tag{7}$$

*Proof.* We set $X \in \mathcal{X}$. Using Perez-Lebel et al. (2025)'s corollary B.4, we have :

$$\mathcal{R}_f(X) = EC(\delta_{f,t}, X) - EC(\delta_{f^*,t^*}, X) \tag{41}$$
$$= \Lambda_\Delta(\delta_{f^*,t^*}(X) - \delta_{f,t}(X))(f^*(X) - t^*) \tag{42}$$

We proceed by case disjunction:

- Case 1: $\delta_{f^*,t^*}(X) = 1$ and $\delta_{f,t}(X) = 0$

$$\mathcal{R}_f(X) = \Lambda_\Delta(\delta_{f^*,t^*}(X) - \delta_{f,t}(X))(f^*(X) - t^*) \tag{43}$$
$$= \Lambda_\Delta(f^*(X) - t^*) \tag{44}$$
$$= \Lambda_\Delta|f^*(X) - t^*| \tag{45}$$

Because $f^*(X) - t^* \geq 0$ since $\delta_{f^*,t^*}(X) = 1$.

- Case 2: $\delta_{f^*,t^*}(X) = 0$ and $\delta_{f,t}(X) = 1$

$$\mathcal{R}_f(X) = \Lambda_\Delta(\delta_{f^*,t^*}(X) - \delta_{f,t}(X))(f^*(X) - t^*) \tag{46}$$
$$= \Lambda_\Delta(t^* - f^*(X)) \tag{47}$$
$$= \Lambda_\Delta|f^*(X) - t^*| \tag{48}$$

Because $f^*(X) - t^* < 0$ since $\delta_{f^*,t^*}(X) = 0$.

- Case 3: $\delta_{f^*,t^*}(X) = \delta_{f,t}(X)$

$$\mathcal{R}_f(X) = \Lambda_\Delta (\delta_{f^*,t^*}(X) - \delta_{f,t}(X))(f^*(X) - t^*) \tag{49}$$
$$= 0. \tag{50}$$

This gives

$$\mathcal{R}_f(X) = \begin{cases} \Lambda_\Delta \mid f^*(X) - t^* \mid & \text{if } \mathbf{1}_{f^*(X) \geq t^*} \neq \mathbf{1}_{f(X) \geq t} \\ 0 & \text{else.} \end{cases} \tag{51}$$

And concludes the proof. □

### A.5  PROOF OF PROPOSITION 6: CONTROL OF THE EPISTEMIC RISK ESTIMATOR

We now prove lemma 3, which will be useful to prove propositions 6 and 7.

For any partitioning estimate $g : X \mapsto \sum_{j=1}^{J} g_j \mathbf{1}_{X \in \mathcal{L}_j}$, define:

$$\mathcal{R}_{f,g}(X) \stackrel{\text{def}}{=} \begin{cases} \Lambda_\Delta \mid c \circ f(X) + g(X) - t^* \mid & \text{if } \mathbf{1}_{c \circ f(X) + g(X) \geq t^*} \neq \mathbf{1}_{f(X) \geq t} \\ 0 & \text{else.} \end{cases} \tag{52}$$

Moreover, let:

$$\Delta \mathcal{R}_f(g, X) \stackrel{\text{def}}{=} \left| \mathcal{R}_f(X) - \mathcal{R}_{f,g}(X) \right|.$$

**Lemma 3.** *For any partitioning estimate* $g : X \mapsto \sum_{j=1}^{J} g_j \mathbf{1}_{X \in \mathcal{L}_j}$ *we have:*

$$\forall j \in [1, J], X \in \mathcal{L}_j, \quad \Delta \mathcal{R}_f(g, X) \leq |\Lambda_\Delta| |r(X) - g_j|.$$

*Or equivalently,*

$$\forall X \in \mathcal{X}, \quad \Delta \mathcal{R}_f(g, X) \leq |\Lambda_\Delta| |r(X) - g(X)|.$$

*Proof.* Let $j \in [1, J]$ and $X \in \mathcal{L}_j$.

We proceed by case disjunction:

- **Case 1:** $f(X) < t$

- Case 1.1: $f(X) < t, f^*(X) \geq t^*, c \circ f(X) + g_j \geq t^*$

We have:
$$\mathcal{R}_f(X) = \Lambda_\Delta \mid f^*(X) - t^* \mid$$

and
$$\mathcal{R}_{f,g}(X) = \Lambda_\Delta \mid c \circ f(X) + g_j - t^* \mid .$$

Therefore,

$$\Delta \mathcal{R}_f(g, X) = \left| \Lambda_\Delta \mid f^*(X) - t^* \mid -\Lambda_\Delta \mid c \circ f(X) + g_j - t^* \mid \right| \tag{53}$$

$$= |\Lambda_\Delta| \times \left| |f^*(X) - t^*| - |c \circ f(X) + g_j - t^*| \right| \tag{54}$$

$$= |\Lambda_\Delta| \left| f^*(X) - t^* - (c \circ f(X) + g_j - t^*) \right| \tag{55}$$

$$= |\Lambda_\Delta| \left| f^*(X) - (c \circ f(X) + g_j) \right| \tag{56}$$

$$= |\Lambda_\Delta| \left| r(X) - g_j \right| \tag{57}$$

(53) By definition

(55) Because $f^*(X) \geq t^*$ and $c \circ f(X) + g_j \geq t^*$

(57) By definition of $r(X)$

- Case 1.2: $f(X) < t, f^*(X) \geq t^*, c \circ f(X) + g_j < t^*$

We have:
$$\mathcal{R}_f(X) = \Lambda_\Delta \mid f^*(X) - t^* \mid$$

and
$$\mathcal{R}_{f,g}(X) = 0.$$

Therefore,

$$\Delta\mathcal{R}_f(g, X) = \left| \Lambda_\Delta \mid f^*(X) - t^* \mid \right| \tag{58}$$
$$= |\Lambda_\Delta||f^*(X) - t^*| \tag{59}$$
$$\leq |\Lambda_\Delta|\left(|f^*(X) - t^*| + |c \circ f(X) + g_j - t^*|\right) \tag{60}$$
$$\leq |\Lambda_\Delta|\left(f^*(X) - t^* + t^* - (c \circ f(X) + g_j)\right) \tag{61}$$
$$\leq |\Lambda_\Delta|\left(f^*(X) - (c \circ f(X) + g_j)\right) \tag{62}$$
$$\leq |\Lambda_\Delta|\left|r(X) - g_j\right| \tag{63}$$

(60) $|c \circ f(X) + g_j - t^*| \geq 0$

(61) $f^*(X) \geq t^*$ and $c \circ f(X) + g_j < t^*$

(63) By definition of $r(X)$

- Case 1.3: $f(X) < t, f^*(X) < t^*, c \circ f(X) + g_j \geq t^*$

We have:
$$\mathcal{R}_f(X) = 0$$

and
$$\mathcal{R}_{f,g}(X) = \Lambda_\Delta \mid c \circ f(X) + g_j - t^* \mid .$$

$$\Delta\mathcal{R}_f(g, X) = \left| \Lambda_\Delta \mid c \circ f(X) + g_j - t^* \mid \right| \tag{64}$$
$$= |\Lambda_\Delta||c \circ f(X) + g_j - t^*| \tag{65}$$
$$\leq |\Lambda_\Delta|\left(|c \circ f(X) + g_j - t^*| + |f^*(X) - t^*|\right) \tag{66}$$
$$\leq |\Lambda_\Delta|\left(c \circ f(X) + g_j - t^* + t^* - f^*(X)\right) \tag{67}$$
$$\leq |\Lambda_\Delta|\left(c \circ f(X) + g_j - f^*(X)\right) \tag{68}$$
$$\leq |\Lambda_\Delta|\left|r(X) - g_j\right| \tag{69}$$

(66) $|f^*(X) - t^*| \geq 0$

(67) $f^*(X) < t^*$ and $c \circ f(X) + g_j \geq t^*$

(69) By definition of $r(X)$

- Case 1.4: $f(X) < t, f^*(X) < t^*, c \circ f(X) + g_j < t^*$

We have:

$$\mathcal{R}_f(X) = 0$$

and

$$\mathcal{R}_{f,g}(X) = 0.$$

Therefore,

$$\Delta\mathcal{R}_f(X) = 0 \tag{70}$$

$$\leq |\Lambda_\Delta|\Big|r(X) - g_j\Big| \tag{71}$$

- **Case 2:** $f(X) \geq t$

- Case 2.1: $f(X) \geq t, f^*(X) < t^*, c \circ f(X) + g_j < t^*$

We have:

$$\mathcal{R}_f(X) = \Lambda_\Delta \mid f^*(X) - t^* \mid$$

and

$$\mathcal{R}_{f,g}(X) = \Lambda_\Delta \mid c \circ f(X) + g_j - t^* \mid .$$

Therefore,

$$\Delta\mathcal{R}_f(g, X) = \Big|\Lambda_\Delta \mid f^*(X) - t^* \mid -\Lambda_\Delta \mid c \circ f(X) + g_j - t^* \mid \Big| \tag{72}$$

$$= |\Lambda_\Delta| \times \Big||f^*(X) - t^*| - |c \circ f(X) + g_j - t^*|\Big| \tag{73}$$

$$= |\Lambda_\Delta|\Big|t^* - f^*(X) - (t^* - (c \circ f(X) + g_j))\Big| \tag{74}$$

$$= |\Lambda_\Delta|\Big|c \circ f(X) + g_j - f^*(X)\Big| \tag{75}$$

$$= |\Lambda_\Delta|\Big|r(X) - g_j\Big| \tag{76}$$

(74) $f^*(X) < t^*$ and $c \circ f(X) + g_j < t^*$

- Case 2.2: $f(X) \geq t, f^*(X) < t^*, c \circ f(X) + g_j \geq t^*$

We have:

$$\mathcal{R}_f(X) = \Lambda_\Delta \mid f^*(X) - t^* \mid$$

and

$$\mathcal{R}_{f,g}(X) = 0.$$

Therefore,

$$\Delta\mathcal{R}_f(g, X) = \Big|\Lambda_\Delta \mid f^*(X) - t^* \mid \Big| \tag{77}$$

$$= |\Lambda_\Delta||f^*(X) - t^*| \tag{78}$$

$$\leq |\Lambda_\Delta|\big(|f^*(X) - t^*| + |c \circ f(X) + g_j - t^*|\big) \tag{79}$$

$$\leq |\Lambda_\Delta|\big(t^* - f^*(X) + (c \circ f(X) + g_j - t^*)\big) \tag{80}$$

$$\leq |\Lambda_\Delta|\big(c \circ f(X) + g_j - f^*(X)\big) \tag{81}$$

$$\leq |\Lambda_\Delta|\Big|r(X) - g_j\Big| \tag{82}$$

(80) $f^*(X) < t^*$ and $c \circ f(X) + g_j \geq t^*$

- Case 2.3: $f(X) \geq t, f^*(X) \geq t^*, c \circ f(X) + g_j < t^*$

We have:
$$\mathcal{R}_f(X) = 0$$

and
$$\mathcal{R}_{f,g}(X) = \Lambda_\Delta \mid c \circ f(X) + g_j - t^* \mid .$$

$$\Delta \mathcal{R}_f(g, X) = \left| \Lambda_\Delta \mid c \circ f(X) + g_j - t^* \mid \right| \tag{83}$$
$$= |\Lambda_\Delta||c \circ f(X) + g_j - t^*| \tag{84}$$
$$\leq |\Lambda_\Delta|\left(|c \circ f(X) + g_j - t^*| + |f^*(X) - t^*|\right) \tag{85}$$
$$\leq |\Lambda_\Delta|\left|c \circ f(X) + g_j - t^* + t^* - f^*(X)\right| \tag{86}$$
$$\leq |\Lambda_\Delta|\left|c \circ f(X) + g_j - f^*(X)\right| \tag{87}$$
$$\leq |\Lambda_\Delta|\left|r(X) - g_j\right| \tag{88}$$

(86) $f^*(X) < t^*$ and $c \circ f(X) + g_j \geq t^*$

- Case 2.4: $f(X) \geq t, f^*(X) < t^*, c \circ f(X) + g_j < t^*$

We have:
$$\mathcal{R}_f(X) = 0$$

and
$$\mathcal{R}_{f,g}(X) = 0.$$

Therefore,
$$\Delta \mathcal{R}_f(g, X) = 0 \tag{89}$$
$$\leq |\Lambda_\Delta|\left|r(X) - g_j\right| \tag{90}$$

This ends the case disjunction and concludes the proof of lemma 3. $\qquad \square$

**Proposition 6** (A Partition-Based Pointwise estimation of the Epistemic decision Risk, A.5). *Let $\hat{r}(X)$ be a partitioning estimate (corresponding to partition $\mathcal{L}$) applied to $R = Y - c \circ f(X)$, and let $t^*$ and $\Lambda_\Delta$ as defined in Section 2.3. We define our estimator:*

$$\widehat{\mathcal{R}}_{f,\mathcal{L}}(X) \overset{def}{=} \begin{cases} \Lambda_\Delta \mid c \circ f(X) + \hat{r}(X) - t^* \mid & \text{if } \mathbf{1}_{c \circ f(X)) + \hat{r}(X) \geq t^*} \neq \mathbf{1}_{f(X) \geq t} \\ 0 & \text{else.} \end{cases} \tag{8}$$

*Then, for any $X$ belonging to region $j \in [1, J]$:*

$$\left|\mathcal{R}_f(X) - \widehat{\mathcal{R}}_{f,\mathcal{L}}(X)\right| \leq |\Lambda_\Delta|\left|r(X) - \hat{r}_j\right|$$

*Proof.* Let $\mathcal{L}$ be a partition of the input space, and $X \in \mathcal{X}$ belonging to region $\mathcal{L}_j$.

From lemma 3, by using $g = \hat{r}$ we have by definition $\widehat{\mathcal{R}}_{f,\mathcal{L}} = \mathcal{R}_{f,\hat{r}}$ and:

$$\Delta \mathcal{R}_f(\hat{r}, X) \leq |\Lambda_\Delta|\left|r(X) - \hat{r}_j\right|$$

which ends the proof of proposition 6.

While proposition 6 provides a pointwise control, it is easy to obtain a control on average over each region of the partition.

From lemma 3, by using $g = r_{\mathcal{L}}^*$ we have :

$$\Delta \mathcal{R}_f(r_{\mathcal{L}}^*, X) \leq |\Lambda_\Delta| \Big| r(X) - r_j^* \Big|$$

Taking the expectation conditional to $\mathcal{L}_j$:

$$\mathbb{E}[\Delta \mathcal{R}_f(X) \mid X \in \mathcal{L}_j] \leq \mathbb{E}\Big[ |\Lambda_\Delta| \Big| r(X) - r_j^* \Big| \mid X \in \mathcal{L}_j \Big] \tag{91}$$

$$\leq |\Lambda_\Delta| \sqrt{\mathbb{E}[(r(X) - r_j^*)^2 \mid \mathcal{L}_j]} \tag{92}$$

$$\leq |\Lambda_\Delta| \sqrt{\mathbb{V}[r(X) \mid \mathcal{L}_j]} \tag{93}$$

(92) Cauchy Schwarz

(93) $r_j^* = \mathbb{E}[Y - c \circ f(X) \mid \mathcal{L}_j] = \mathbb{E}[f^*(X) - c \circ f(X) \mid \mathcal{L}_j] = \mathbb{E}[r(X) \mid \mathcal{L}_j]$

$\square$

## A.6   PROOF OF PROPOSITION 7: CONSISTENCY OF THE EPISTEMIC RISK ESTIMATOR

**Proposition 7** (Consistency of the epistemic decision risk estimator, A.6). *Let $\hat{r}^{(n)}$ be a partitioning estimate applied to $R = Y - c \circ f(X)$. Assume that $\hat{r}^{(n)}$ is weakly universally consistent in $L_2$. Then the sequence of estimators $\widehat{\mathcal{R}}_{f,\mathcal{L}^{(n)}}^{(n)}$ is weakly universally consistent in $L_2$, i.e.,*

$$\mathbb{E}_{\mathcal{D}^{(n)}} \Big[ \mathbb{E}_X \big[ \, | \, \widehat{\mathcal{R}}_{f,\mathcal{L}^{(n)}}^{(n)}(X) - \mathcal{R}_f(X) \, |^2 \, \big] \Big] \xrightarrow[n \to +\infty]{} 0.$$

*Proof.* We have :

$$\mathbb{E}_{\mathcal{D}^{(n)}} \Big[ \mathbb{E}_X \big[ \, | \, \mathcal{R}_f(X) - \widehat{\mathcal{R}}_{f,\mathcal{L}^{(n)}}^{(n)}(X) \, |^2 \, \big] \Big] \leq \mathbb{E}_{\mathcal{D}^{(n)}} [\mathbb{E}_X [\Lambda_\Delta^2 (r(X) - \hat{r}^{(n)}(X))^2]] \tag{94}$$

$$\leq \Lambda_\Delta^2 \mathbb{E}_{\mathcal{D}^{(n)}} [\mathbb{E}_X [(r(X) - \hat{r}^{(n)}(X))^2]] \tag{95}$$

$$\xrightarrow[n \to +\infty]{} 0 \tag{96}$$

(95) Lemma 3 applied to $g = \hat{r}^{(n)}$

(96) Consistency of $\hat{r}^{(n)}$

Which concludes the proof.

$\square$

## A.7   HONEST TREES CONSISTENCY

We recall that honest trees proceed in two stages:

1. Learn the tree partition (i.e., the splits) using one subset of the data.
2. Estimate the leaf values using a disjoint subset.

Because disjoint data subsets are used for partitioning and leaf value estimation, the analysis is greatly simplified. In particular, the partition can be treated as fixed when analyzing the leaf value estimates, since it does not depend on the data used for estimation. It allows to directly apply Theorem 4.2 from Györfi et al. (2002).

We consider that the number of samples used for learning the partition (training set) and the number of samples used for leaf values estimation (evaluation set) *both* increase with the total number of samples available, so that both tend to infinity when the total number of samples tend to infinity.

Based on previous results (Meinshausen, 2006; Györfi et al., 2002), we can show that honest trees satisfying the following 3 assumptions are consistent.

**Assumption 1:** $X$ is a random variable with compactly supported distribution.

*Note:* In Meinshausen (2006), it is assumed that features are bounded in $[0,1]^d$ for notational purposes, but the distribution of $X$ only needs to be of compact support.

**Assumption 2:** The proportion of observations in a leaf $j$ relative to all observations is vanishing for large $n$, *i.e.* $\max_{j \in [1, J^{(n)}]} n_j / n \xrightarrow[n \to +\infty]{} 0$. The minimal number of observations in a leaf is growing for large $n$, that is $1/\min_{j \in [1, J^{(n)}]} n_j \xrightarrow[n \to +\infty]{} 0$.

**Assumption 3:** The probability that variable $m = 1, ..., d$ is chosen to construct a split point is bounded from below for every leaf by a positive constant. After a split, child leaves contain at least a proportion $\gamma$ of the parent leaf observations, for some $0 < \gamma \le 0.5$.

Meinshausen (2006)'s Lemma 2 shows that for trees following the above construction assumptions, the leaves' sizes converge to 0 as the number of samples in the training set grows (equivalently as the number of total samples grows), within the limits of the support of $X$. Formally, this means that

$$\max_{j \in [1, J^{(n)}], \mathcal{L}_j \cap supp(X) \neq \emptyset} diam(\mathcal{L}_j) \xrightarrow[n \to +\infty]{} 0.$$

As we show below, the second part of assumption 2 implies that the total number of leaves relative to the number of samples available in the evaluation set is vanishing (i.e. $\frac{J^{(n)}}{n} \xrightarrow[n \to +\infty]{} 0$), for leaves within the support of $X$.

Let $n$ be the total number of samples of the evaluation set, and let $n_j$ denote the number of samples in leaf $j$, $j = 1, \ldots, J^{(n)}$. Then

$$n = \sum_{j=1}^{J^{(n)}} n_j \ge J^{(n)} \min_{1 \le j \le J^{(n)}} n_j \quad \implies \quad \frac{n}{J^{(n)}} \ge \min_{1 \le j \le J^{(n)}} n_j.$$

Equivalently,

$$0 \le \frac{J^{(n)}}{n} \le \frac{1}{\min_{1 \le j \le J^{(n)}} n_j}.$$

If Assumption 2 (part 2) is verified, it follows that

$$\frac{J^{(n)}}{n} \longrightarrow 0 \quad \text{as } n \to \infty.$$

Therefore, we can apply Györfi et al. (2002)'s Theorem 4.2.

**Theorem** (Theorem 4.2 of Györfi et al. (2002)). *If for each sphere $S$ centered at the origin*

$$\max_{j \in [1, J^{(n)}], \mathcal{L}_j \cap S \neq \emptyset} diam(\mathcal{L}_j) \xrightarrow[n \to +\infty]{} 0$$

*and*

$$\frac{|\{j : \mathcal{L}_j \cap S \neq \emptyset\}|}{n} \xrightarrow[n \to +\infty]{} 0$$

*then the partitioning regression function estimate is weakly universally consistent in $L_2$.*

Hence, honest trees constructed based on Meinshausen (2006)'s assumptions lead to consistent partitioning estimates.

## B ADDITIONAL EXPERIMENTAL DETAILS

### B.1 RECALL OF THE BIN-WISE CALIBRATION RISK

**Proposition 8** (Calibration decision risk, Perez-Lebel et al. (2025)). *For any confidence score $f(x) \in [0,1]$, the binwise calibration risk writes:*

$$\mathcal{R}_f^{CL}(f(x)) = \begin{cases} \Lambda_\Delta \mid c \circ f(x) - t^* \mid & \textit{if } \mathbf{1}_{c \circ f(x) \ge t^*} \neq \mathbf{1}_{f(x) \ge t} \\ 0 & \textit{else.} \end{cases} \tag{97}$$

## B.2 HONEST TREE CONSTRUCTION ALGORITHM

We describe here the algorithm used to compute our partitioning estimate from an honest tree.

---

**Algorithm 1** Honest tree $\hat{r}$

---

**Input**: Probabilistic classifier $f$, $(X_{cal}, Y_{cal})$, $(X_{fit}, Y_{fit})$, $(X_{eval}, Y_{eval})$
**Fit** $c \circ f$ using Platt scaling with $(X_{cal}, Y_{cal})$
**Fit** the tree $\hat{r}$ on $(X_{fit}, Y_{fit} - c \circ f(X_{fit}))$      ▷ We now have the partition $\mathcal{L}$ from $\hat{r}$
**for** $\mathcal{L}_j \in \mathcal{L}$ **do**
    **Compute** $\hat{r}_j = \frac{\sum_{i \in eval} \mathbf{1}_{X_i \in \mathcal{L}_j}(Y_i - c \circ f(X_i))}{\sum_{i \in eval} \mathbf{1}_{X_i \in \mathcal{L}_j}}$
**end for**

---

## B.3 DISCUSSION ON HYPERPARAMETERS AND IMPLEMENTATION CHOICES

**Calibration** - We use Platt scaling (Platt, 1999) to obtain $c \circ f$. The quality of calibration matters: if the underlying model is poorly calibrated, our approach will partly capture miscalibration error. For Figure 3, we allocate at least 4000 samples to the calibration set, or follow the proportions described in 3.3 to limit the variance at low sample sizes due to a bad calibration $c$.

**Hyperparameters** - We adopt a conservative strategy in our experiments: we use a single fixed set of hyperparameters for the partitioning estimate $\hat{r}$ across all tasks (unlimited depth with a minimum of 15 samples per leaf at training time). Although task-specific tuning could yield stronger results, this setup illustrates the robustness of our chosen hyperparameters. We split the data into calibration (10%), fitting (40%) and evaluation (50%) sets.

**Choice of input space** - The input space on which the tree is fitted can vary. In standard tabular problems, the natural choice is the original feature space. In contrast, neural networks often operate on less structured data (e.g., images, text) and produce richer internal representations. In such cases, the partition can be built on an embedded representation of the data—for example, the activations of the last hidden layer before the softmax (Perez-Lebel et al., 2023)—or on a learned low-dimensional embedding such as UMAP (Detommaso et al., 2024).

## B.4 SEMI-SYNTHETIC DATA GENERATION.

For fig. 3, we use the Weather dataset from TabReD (Rubachev et al., 2025). We retain the two most important features, as identified by permutation importance (Breiman, 2001). To obtain a semi-synthetic version (i.e., with access to $f^*$), we fit a ground-truth model $f^*(X)$ on half of the test set split. Synthetic class labels are then generated for the remaining half by resampling from $f^*(X)$ according to a Bernoulli distribution, $Y \sim \mathcal{B}(f^*(X))$. This yields a semi-synthetic test dataset where $X$ consists of the original features and $Y$ is sampled from $f^*(X)$.

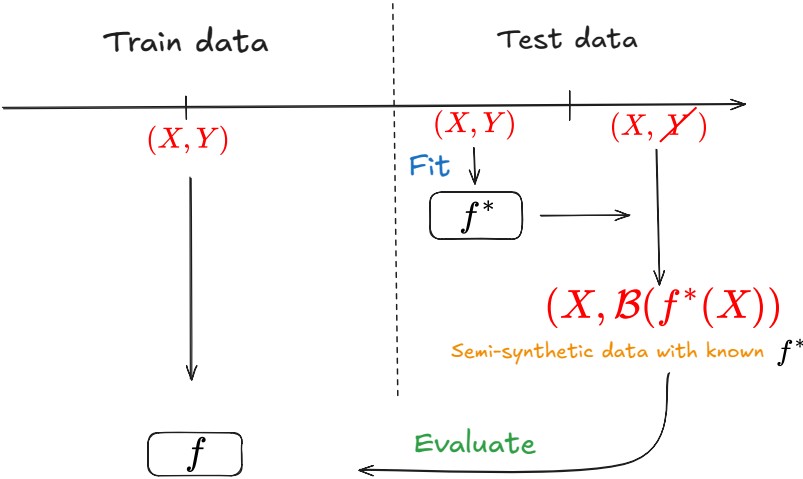

Figure 7: **Diagram of the semi-synthetic dataset construction**

### B.5 LIST OF LLMS USED IN SECTION 4.2 AND 4.3

All models were gathered from the HuggingFace API.

Table 1: List of models used. Models in **bold** were used to create the cascades.

| Model Name | Size | Company | Type |
|---|---|---|---|
| DeepSeek-R1-Distill-Llama | 8B, 70B | DeepSeek AI | Non Instruct |
| **Gemma 2** | 9B, **27B** | Google | **Instruct** & Non Instruct |
| Llama-2 | 13B, 70B | Meta | Instruct & Non Instruct |
| Llama-3 | 8B, 70B | Meta | Instruct & Non Instruct |
| **Llama-3.1** | **8B** | Meta | **Instruct** & Non Instruct |
| **Llama-3.2** | **1B, 3B** | Meta | **Instruct** & Non Instruct |
| **Llama-3.3** | **70B** | Meta | **Instruct** |
| Mistral | 7B (v0.3) | Mistral AI | Instruct & Non Instruct |
| **Mixtral** | **8x7B** (v0.1) | Mistral AI | **Instruct** & Non Instruct |
| Orca-2 | 13B | Microsoft | Non Instruct |
| **phi-4** | **14B** | Microsoft | **Non Instruct** |

### B.6 DIAGRAM OF OUR CASCADING METHOD

We give a visualization of our cascading method. We consider a set of LLMs, ordered from smallest to biggest in size. The cascade starts with the smallest LLM and proceeds as follows:

1. Send the question to the LLM and recover its confidence $f(X)$.

2. Verify if the estimated risk $\widehat{\mathcal{R}}_f(X)$ is smaller than the fixed threshold $\tau$.

3. If yes, stop the cascade and use the answer of the model.

4. If not, take the next model in the cascade and go back to step 1.

If no model in the cascade has a risk estimate below $\tau$, we return the answer of the model with the lowest estimated risk.

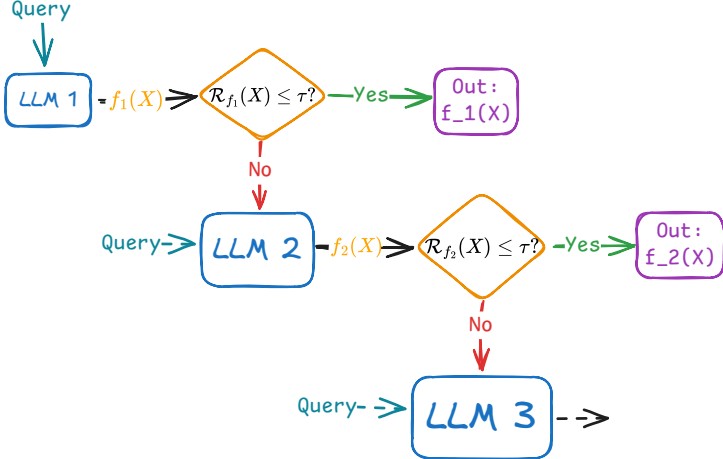

Figure 8: Diagram of our cascading method

## C  ADDITIONAL EXPERIMENTS

### C.1  QUANTIFYING SUBOPTIMALITY

The expected excess decision risk provides a principled criterion for evaluating whether a model would benefit from fine-tuning or other forms of post-training.

We replicate the hate speech detection tasks experiments of Perez-Lebel et al. (2025), which aim to evaluate the correlation between estimated excess decision cost and actual decision cost reduction after post-training.

### C.1.1  HATE SPEECH EXPERIMENTAL SETUP

The experimental setup is briefly recalled here, see Perez-Lebel et al. (2025) for details.

Hate speech detection consists in a binary classification with $Y=1$ for hate speech and $Y=0$ otherwise. The study is conducted across 14 datasets (listed in section C.1.3) and 6 pretrained language models (listed in section C.1.2) which provide confidence scores to be evaluated.

The experiments use the feature embeddings from each model's penultimate layer as the input for all post-training methods.

Experimental Procedure:

1. The text from all 14 datasets is processed by each of the 6 pre-trained models. For every sample, the following triplet is recorded: (i) the model's predicted probability $f(X)$; (ii) the model's internal feature embedding $X$; (iii) the ground-truth label $Y$.

2. The collected data triplets are partitioned into a *train set* and a *test set*.

3. The post-training method are fitted on the *train set*.

4. Using only the *test set* data (prior to any corrections), the excess decision risk estimators are calculated.

5. The post-training corrections learned in the previous step are applied to the *test set*. The actual gain in expected utility is measured by comparing the performance of the corrected model against the original, uncorrected model, or between different post-training methods.

### C.1.2 MODELS USED

Table 2: Detailed description of all the models used for the quantifying suboptimality experiment.

| Model | HuggingFace | Latent Layer |
|---|---|---|
| CNERG Hatexplain | Hate-speech-CNERG/bert-base-uncased-hate... | classifier |
| CNERG en MURIL | Hate-speech-CNERG/english-abusive-MuRIL | classifier |
| CNERG en mono | Hate-speech-CNERG/dehatebert-mono-englis... | classifier |
| CNERG portuguese | Hate-speech-CNERG/dehatebert-mono-portug... | classifier |
| CNERG tamil | Hate-speech-CNERG/tamil-codemixed-abusiv... | classifier |
| FB Roberta | facebook/roberta-hate-speech-dynabench-r... | classifier.out_proj |

### C.1.3 DATASETS USED

Table 3: Detailed description of all the datasets used for the quantifying suboptimality experiment.

| Dataset | HuggingFace or CSV | Split | Input | Target | Pos. class |
|---|---|---|---|---|---|
| Tweets | tweets_hate:. | train | tweet | label | |
| Speech18 | hate_speech18 | train | text | label | |
| Offensive | hate_speech_offensive | train | tweet | class | 0 |
| Davidson | krishan-CSE/... | train+test | text | labels | 0 |
| Gender | ctoraman/... | train+test | Text | Label | 2 |
| FRENK | classla/FRENK-hate-en | train+val+test | text | label | |
| | limjiayi/hatef.. | train+val+test | text | label | |
| Check | Paul/hatecheck | test | test_case | label_gold | hateful |
| Tweets 2 | thefrankhsu/... | train+test | tweet | label | |
| Open | parnoux/hate_... | test | tweet | class | 0 |
| UCB | ucberkeley-dl... | train | text | hate_speec... | y > 0.5 |
| Merged | tweets_hat.. | train | tweet | label | |
| | hate_speech18 | train | text | label | |
| | hate_speech_offensive | train | tweet | class | 0 |
| | krishan-CSE/... | train+test | text | labels | 0 |
| | ctoraman/gende... | train+test | Text | Label | 2 |
| Merged 2 | classla/FRENK-hate-en | train+val+test | text | label | |
| | limjiayi/hatef... | train+val+test | text | label | |
| | Paul/hatecheck | test | test_case | label_gold | hateful |
| | thefrankhsu/hat... | train+test | tweet | label | |
| | parnoux/hat... | test | tweet | class | 0 |
| DynGen | CSV: bvidgen/Dynamic... | | text | label | hate |
| Merged3 | tweets:. | train | tweet | label | |
| | hate_speech18 | train | text | label | |
| | hate_speech_offensive | train | tweet | class | 0 |
| | krishan-CSE/Da... | train+test | text | labels | 0 |
| | ctoraman/ge... | train+test | Text | Label | 2 |
| | classla/FRENK-hate-en | train+val+test | text | label | |
| | limjiayi/hateful... | train+val+test | text | label | |
| | Paul/hatecheck | test | test_case | label_gold | hateful |
| | thefrankhsu/hate... | train+test | tweet | label | |
| | parnoux/hate:. | test | tweet | class | 0 |

### C.1.4 RESULTS

Perez-Lebel et al. (2025) first compares their grouping decision risk estimator to the cost reduction brought by fine-tuning, after recalibration. Based on our excess decision risk formulation, our own $\widehat{\mathcal{R}}^{GL}$ estimator writes as follows:

**Definition 4** (Grouping decision risk estimator). *For $X \in \mathcal{X}$ the pointwise grouping risk estimator writes:*

$$\mathcal{R}_f^{\mathrm{GL}}(X) = \begin{cases} \Lambda_\Delta \mid c \circ f(X) + \hat{r}(X) - t^* \mid & \textit{if } \mathbf{1}_{c \circ f(X) + \hat{r}(X) \geq t^*} \neq \mathbf{1}_{c \circ f(X) \geq t^*} \\ 0 & \textit{else.} \end{cases} \quad (98)$$

We now replicate Perez-Lebel et al. (2025)'s experiments with our estimator instead of theirs. Figure 9 shows the correlation between our grouping decision risk estimator $\mathcal{R}_f^{\mathrm{GL}}$ and the observed fine-tuning gain after calibration. We observe a notable increase in squared Spearman rank correlation, from $r^2 = 0.45$ with Perez-Lebel et al. (2025, fig. 5)'s estimator to $r^2 = 0.61$ (fig. 9) with ours. This demonstrates that our estimator predicts better the benefit of post-training, outperforming the previous estimator, which itself outperformed standard metrics such as accuracy, Brier score, or AUC.

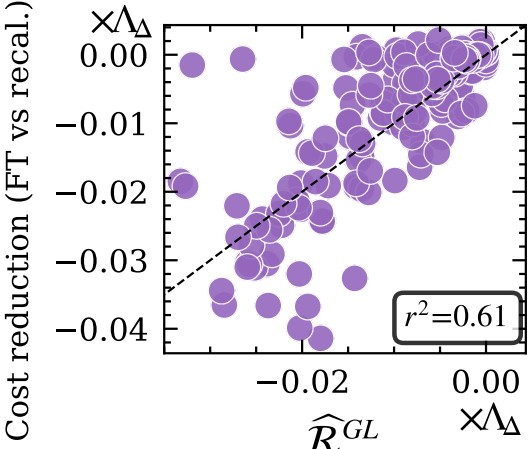

Figure 9: **Correlation between fine-tuning gain after re-calibration, and grouping decision risk estimation.** The y-axis shows the average cost reduction achieved by fine-tuning recalibrated probabilities. Each point corresponds to a (model, dataset, cost matrix $\Lambda$) triplet. The dashed line corresponds to $y = x$.

Furthermore, fig. 10 correlates the total risk with the cost reduction achieved by fine-tuning the base model. We again observe an increase in squared Spearman rank correlation, from $r^2 = 0.83$ in Perez-Lebel et al. (2025)'s figure 6.a to $r^2 = 0.88$ here.

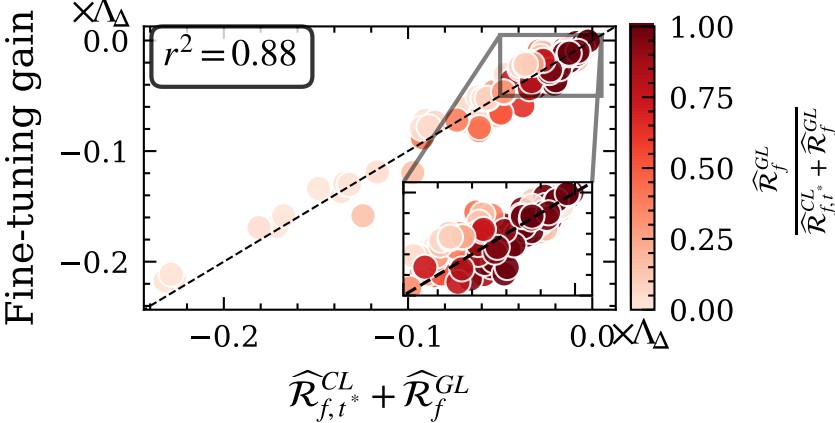

Figure 10: **Correlation between fine-tuning gain and excess decision risk estimation.** The y-axis shows the average cost reduction achieved by fine-tuning the base model. Each point corresponds to a (model, dataset, cost matrix $\Lambda$) triplet. The dashed line corresponds to $y = x$.

We give here the grouping loss evaluations for every ACS dataset separately (figs. 11 to 15). We observe that the grouping loss is consistently non-negligible and generally decreases as model size increases, although its magnitude and rate of decay vary across datasets.

For each LLM, GL estimates were averaged over 5 random seeds governing the calibration, fitting, and evaluation splits. We report the mean and standard deviation per model over these 5 splits. The base and instruct trends were computed using a LOWESS regression.

With our Brier score–based estimator, a grouping loss of 0.01 corresponds to a squared probability error, i.e., an average deviation of 0.1 or less in individual confidence scores (see e.g. group deviations in fig. 1).

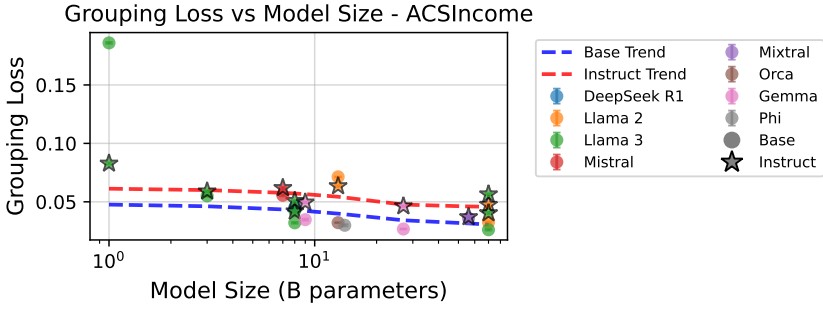

Figure 11: Grouping loss as a function of LLM size, for base and instruct models, ACSIncome.

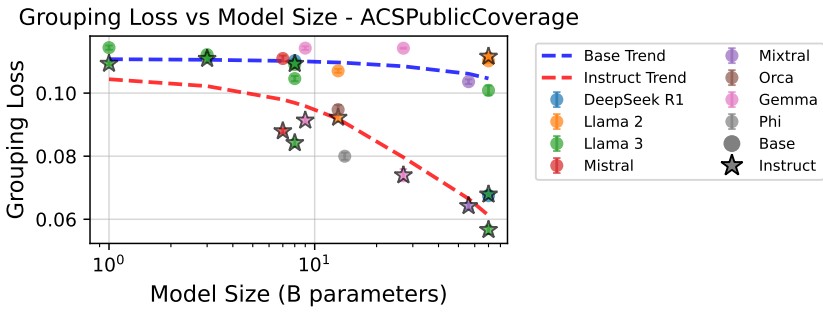

Figure 12: Grouping loss as a function of LLM size, for base and instruct models, ACSPublicCoverage.

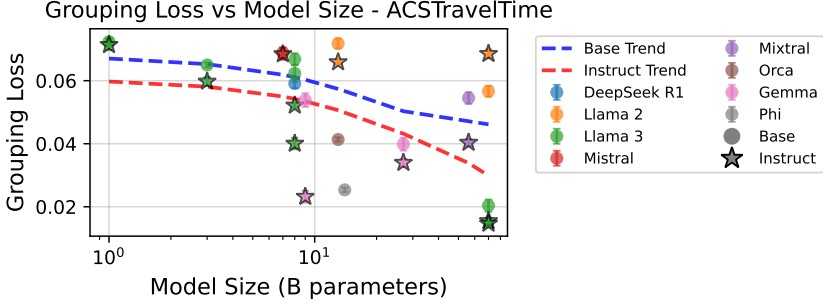

Figure 13: Grouping loss as a function of LLM size, for base and instruct models, ACSTravelTime.

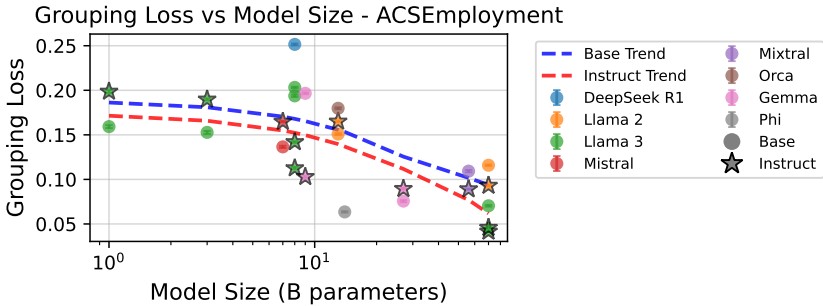

Figure 14: Grouping loss as a function of LLM size, for base and instruct models, ACSEmployment.

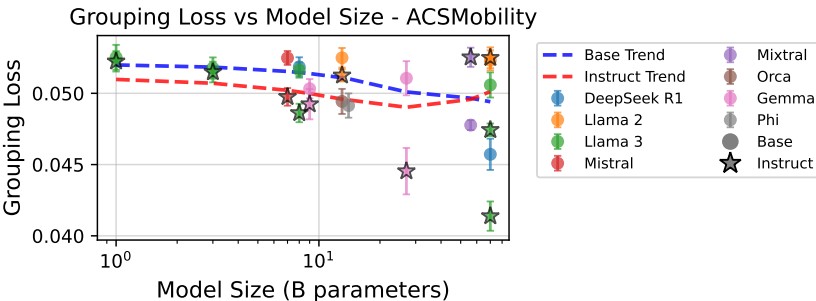

Figure 15: Grouping loss as a function of LLM size, for base and instruct models, ACSMobility.

## C.3 LLM ACCURACY ON EACH ACS DATASET

We report in table 4 LLMs' test accuracies (confidence scores thresholded at $0.5$ as in the cascading experiment) across ACS datasets. Standard deviations are computed across train/test splits. We note that on ACS TravelTime, all models perform close to chance level (the best constant predictor achieves 56.3% accuracy on this dataset) (Ding et al., 2021)).

Table 4: Model Accuracies across ACS Datasets

| Model | ACS Income | ACS Employment | ACS Mobility | ACS PublicCoverage | ACS Travel Time |
|---|---|---|---|---|---|
| Llama-3.2-1B-Instruct | $0.6307 \pm 0.0024$ | $0.4565 \pm 0.0038$ | $0.3426 \pm 0.0043$ | $0.5038 \pm 0.0062$ | $0.4415 \pm 0.0070$ |
| Llama-3.2-3B-Instruct | $0.3920 \pm 0.0013$ | $0.5006 \pm 0.0026$ | $0.7309 \pm 0.0034$ | $0.3787 \pm 0.0026$ | $0.5615 \pm 0.0053$ |
| Llama-3.1-8B-Instruct | $0.6399 \pm 0.0013$ | $0.7184 \pm 0.0021$ | $0.7311 \pm 0.0034$ | $0.7015 \pm 0.0049$ | $0.5337 \pm 0.0056$ |
| phi-4 | $0.7327 \pm 0.0022$ | $0.7734 \pm 0.0018$ | $0.6155 \pm 0.0037$ | $0.2986 \pm 0.0048$ | $0.5773 \pm 0.0062$ |
| gemma-2-27b-it | $0.7378 \pm 0.0023$ | $0.7392 \pm 0.0017$ | $0.7311 \pm 0.0034$ | $0.6262 \pm 0.0033$ | $0.5295 \pm 0.0059$ |
| Mixtral-8x7B-Instruct-v0.1 | $0.7689 \pm 0.0028$ | $0.7390 \pm 0.0013$ | $0.7311 \pm 0.0034$ | $0.7503 \pm 0.0047$ | $0.5136 \pm 0.0063$ |
| Llama-3.3-70B-Instruct | $0.6756 \pm 0.0017$ | $0.7562 \pm 0.0015$ | $0.6834 \pm 0.0040$ | $0.7051 \pm 0.0032$ | $0.5187 \pm 0.0062$ |

## C.4 COMPUTING THE RISKS ON AN EMBEDDING OF THE QUERIES

In this section, we present the results of the cascade when inputs are treated as free text rather than tabular data. For the baseline, we use Hu et al. (2024)'s kNN with the best hyperparameter they found ($k = 40$ with cosine similarity and embedding the queries using the all-MiniLM-L12-v2 embedding model). We do not apply any dimension reduction and keep the embeddings of dimension 384. The embedded queries serve as input space for our partitioning estimate.

Results are shown in fig. 16. For ACSTravelTime, all models perform close to random accuracy in the cascade (see table 4). As a result, models are arbitrarily passed through the cascade, which artificially inflates the costs.

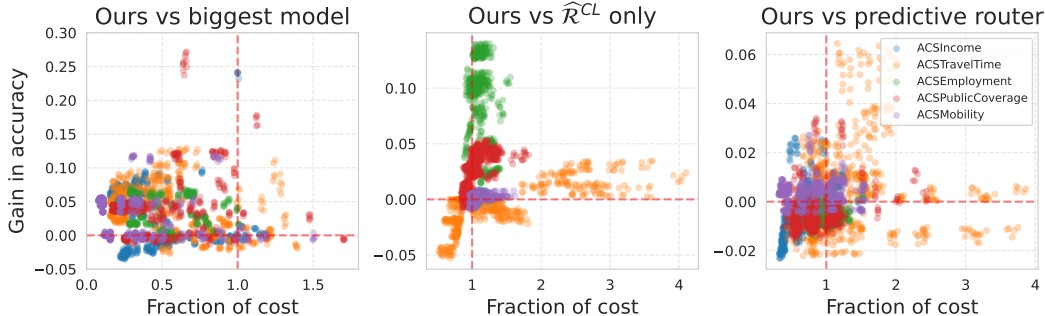

Figure 16: **Cascades: our risk estimator gives better accuracies at a lower cost.** The router and risk estimators are fitted on the embeddings of the prompts given to the LLMs. Each point corresponds to a cascade (an LLM sequence). The y-axis is Acc(ours) - Acc(baseline). Values are positive when our cascade improves accuracy over the baseline. The x-axis is Cost(ours)/Cost(baseline). For values below 1, our model is less expensive. Our cascade based on $\widehat{\mathcal{R}}$ ($\tau = 0$) is compared to **Left:** the biggest LLM in the cascade, **Middle:** A cascade based on $\widehat{\mathcal{R}}^{CL}$ only ($\tau = 0$). **Right:** the predictive router router baseline ($\lambda = 100$).

## C.5 COMPARING TO CASCADES DEFERRING BASED ON CONFIDENCE SCORES.

We evaluate the classical confidence-based cascades, mentioned in section 4.3. We start with the smallest LLM, order them by increasing size, and apply a decision threshold $c > 1/2$. Based on Algorithm 1 in Jitkrittum et al. (2023), the confidence-based cascade proceeds as follows:

1. Send the question to the LLM and recover its confidence $f(X)$.

2. Verify if either $f(X) > c$ or $1 - f(X) > c$ (i.e. if either the confidence for class 0 or 1 exceeds $c$).

3. If yes, stop the cascade and use the anwer of the model.

4. If not, take the next model in the cascade and go back to step 1.

Figures 17 and 18 show that confidence-based cascades perform poorly in our experiments. These poor results most probably stem from the high miscalibration and grouping loss of LLMs. In contrast, our method aims at assessing the quality of confidence scores and accepts models when they incur low excess decision error, not when they are confident enough. In fig. 18, our method achieves a +7% increase in accuracy on average while using only 52% of the cost on average, and is strictly Pareto-optimal in 90% of the cases

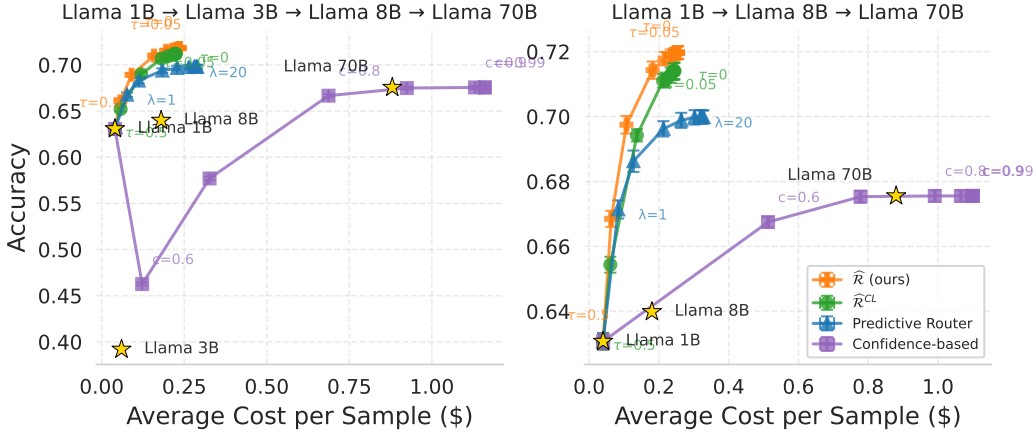

Figure 17: **Cost-accuracy tradeoffs when varying the threshold of risk $\tau$ and willingness to pay $\lambda$ and confidence threshold $c$ on two Llama 3 instruction-tuned family cascades.** Adding the confidence-based traditional approach for cascades.

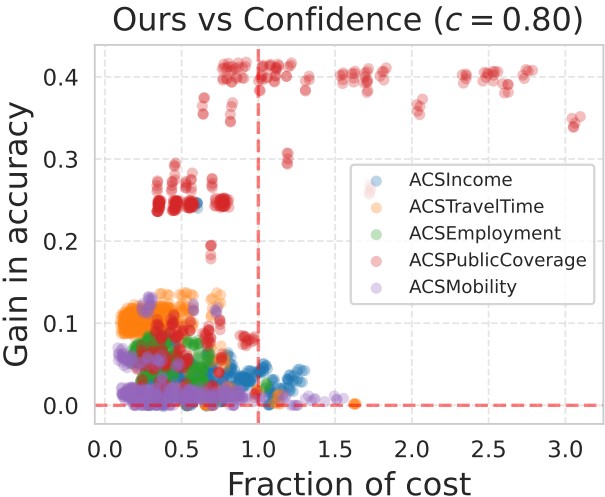

Figure 18: **Accuracy gain versus relative cost for our risk-based cascade compared to the traditional confidence-based cascade.** Confidence threshold set at 0.8 and risk threshold set at 0.

## C.6 TREE DEPTH ABLATION STUDY

We investigate the effect of tree depth on the cascade results described in section 4.3. The experimental setup remains identical, with the only change being the `max_depth` parameter of the honest tree for the partitioning estimate, which we vary over $\{3, 6, 9, 12\}$. Overall, our results are very robust to the `max_depth` value (see figs. 19 to 23). The results of the cascade experiments hold up down to `max_depth = 3`, though the accuracy gain becomes more limited. This is expected as very coarse trees reduce the capacity of $\mathcal{R}$ to improve over $\mathcal{R}^{CL}$ alone.

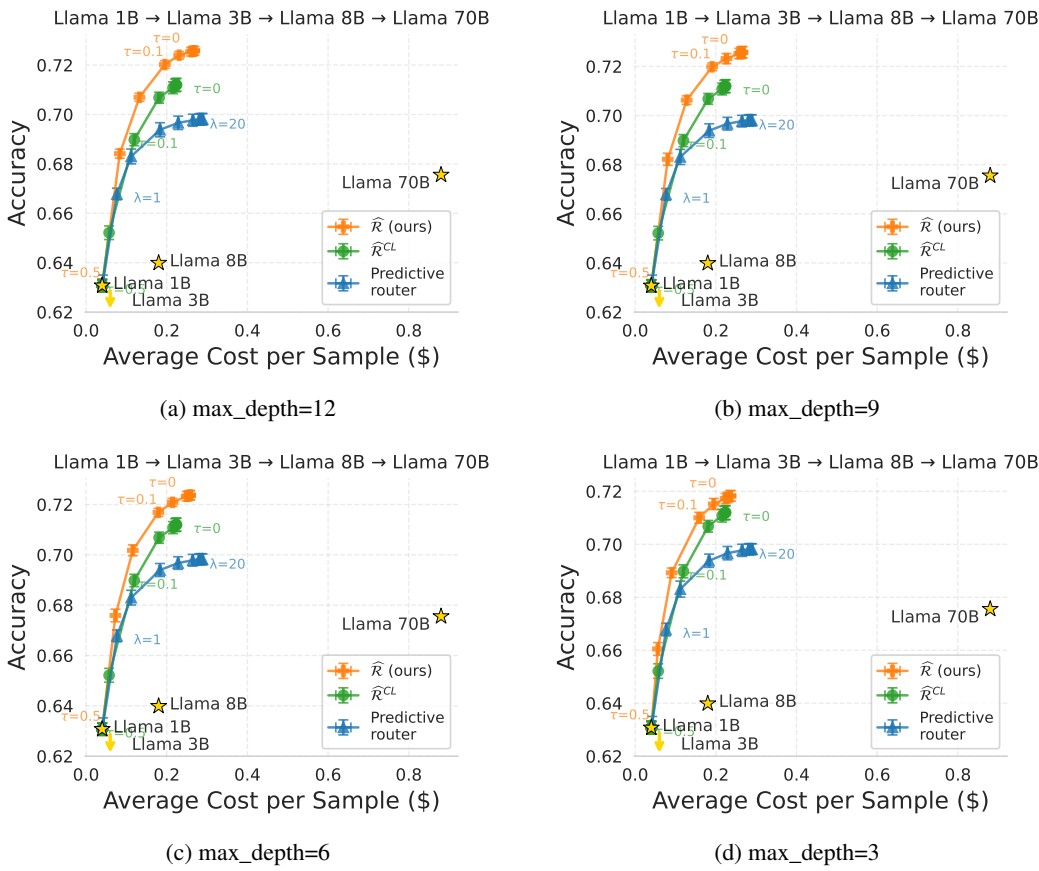

Figure 19: **Cost-accuracy tradeoffs** when varying the threshold of risk $t$ and willingness to pay $\lambda$ on the Llama 3 instruction-tuned family cascade. Changing the max_depth parameter of the honest tree.

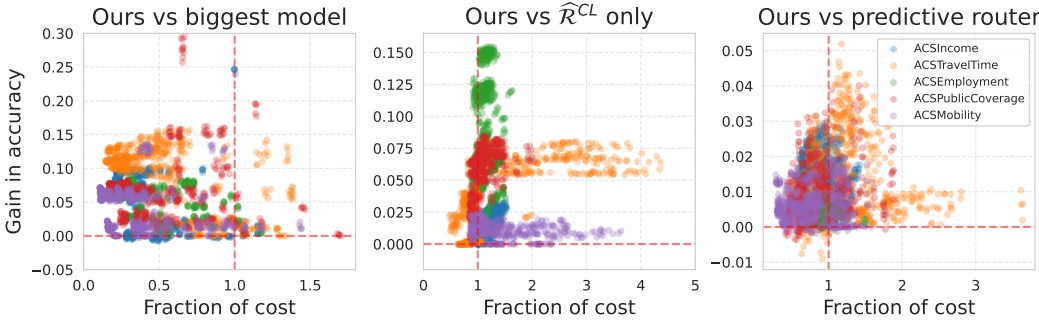

Figure 20: Same experimental setup as section 4.3, max_depth=12 for the honest tree.

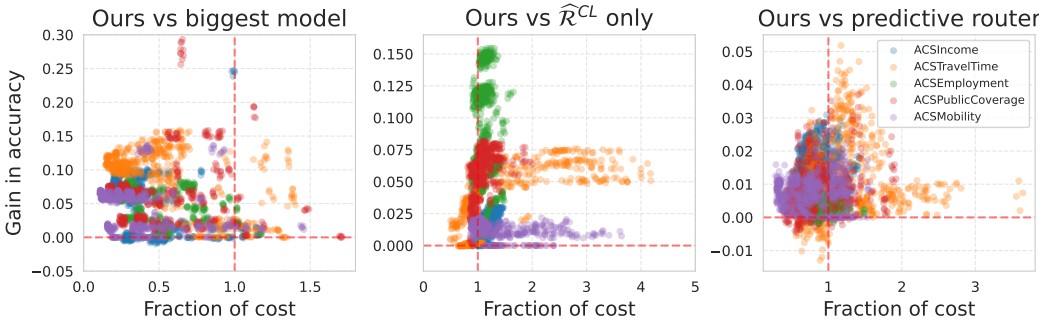

Figure 21: Same experimental setup as section 4.3, max_depth=9 for the honest tree.

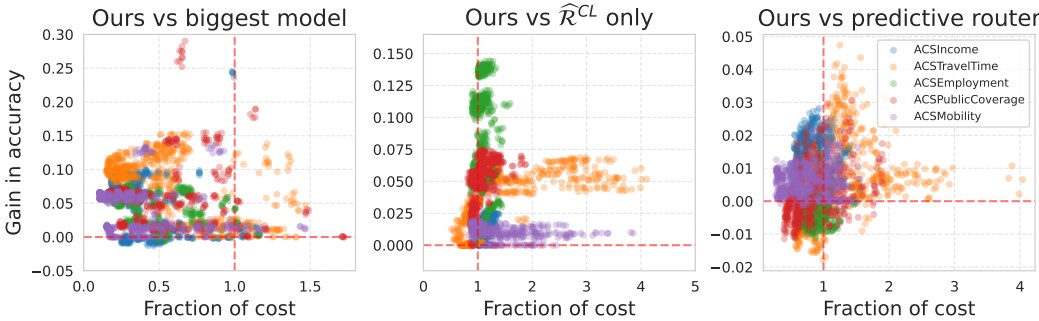

Figure 22: Same experimental setup as section 4.3, max_depth=6 for the honest tree.

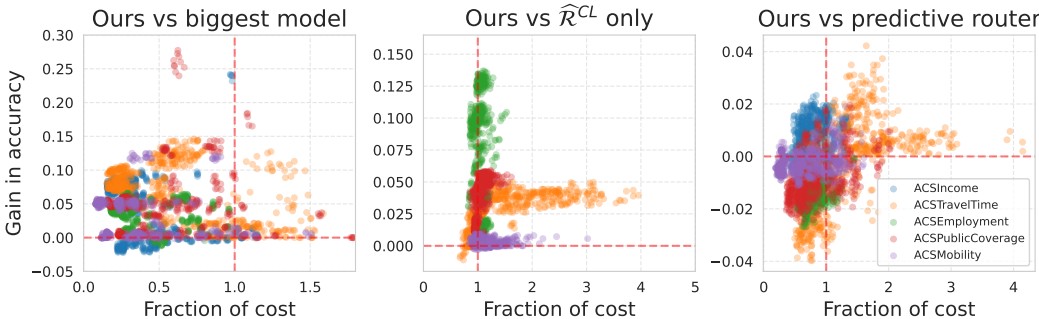

Figure 23: Same experimental setup as section 4.3, max_depth=3 for the honest tree.

## C.7 TRAINING SET SIZE EVALUATION

We vary the number of samples available to train the router and risk estimators (see figs. 24 to 27). For the router, it corresponds to the number of queries labeled with whether or not the LLM was correct. For the calibration risk $\widehat{\mathcal{R}}^{CL}$, it corresponds to the number of samples used to calibrate confidence scores. For our epistemic risk estimator $\widehat{\mathcal{R}}_f$, it corresponds to the number of samples used to calibrate the confidence scores, create the partition from a decision tree and estimate leaf values.

For the ACSTravelTime dataset, all models show around chance accuracy (see table 4).

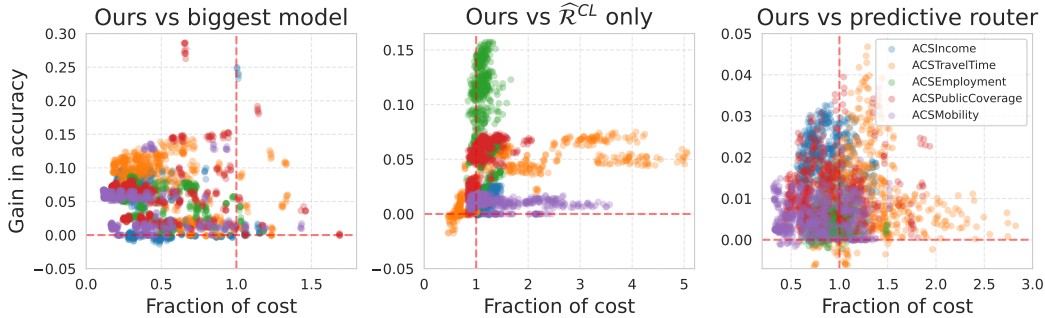

Figure 24: Varying the number of samples used to train the estimators: $n_{train} = 50.000$.

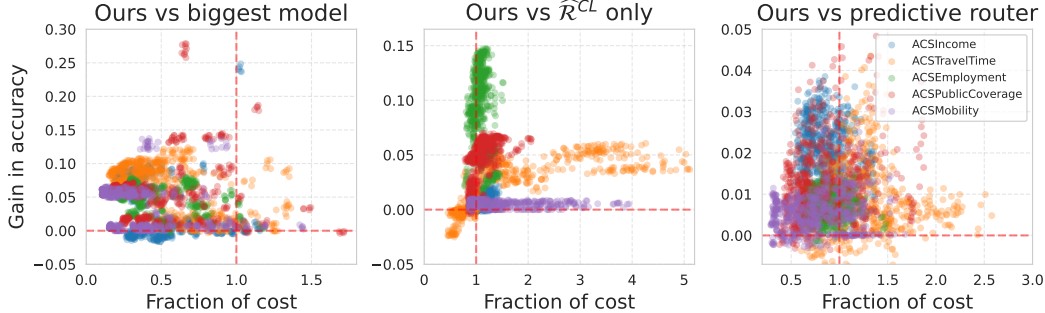

Figure 25: Varying the number of samples used to train the estimators: $n_{train} = 10.000$.

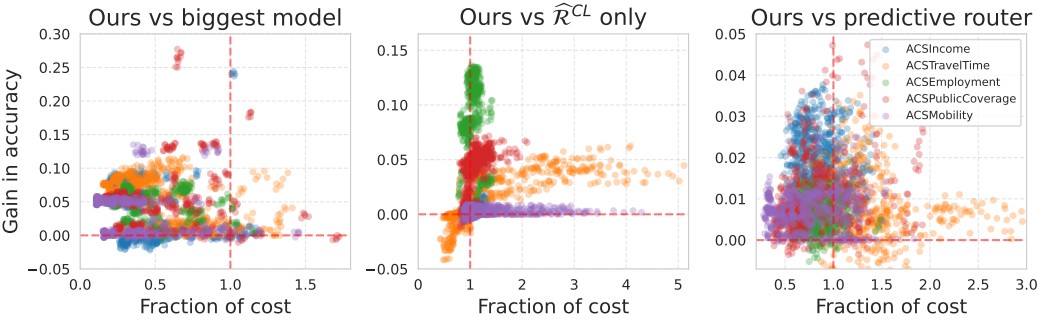

Figure 26: Varying the number of samples used to train the estimators: $n_{train} = 6.000$.

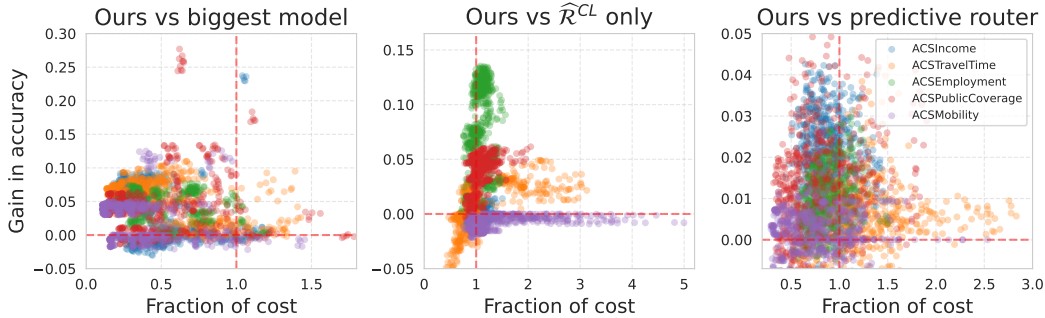

Figure 27: Varying the number of samples used to train the estimators: $n_{train} = 2.000$.

## C.8 USING A STRONGER MODELS TO PREDICT INDIVIDUAL RESIDUALS

We present in this section the results of our cascade experiment using a Histogram Gradient Boosting Regressor for our residual estimate $\hat{r}$. As expected, the accuracy of our approach increases with this method.

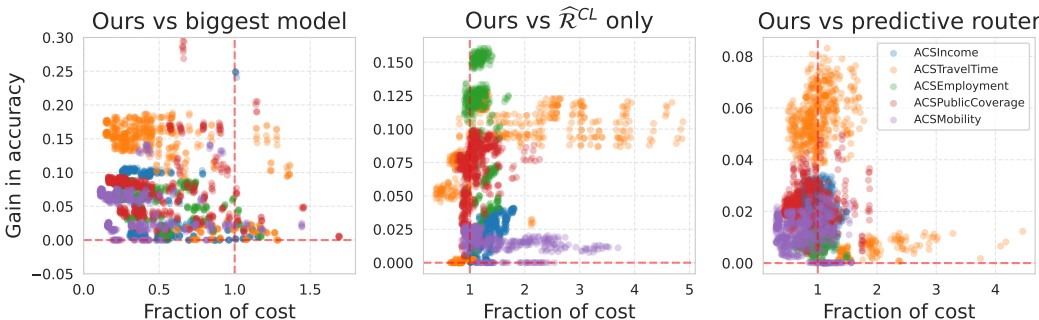

Figure 28: **Using a stronger model for the residual estimate $\hat{r}$ yields higher accuracy gains.** We reproduce the experimental setting of 4.3 using a HistGradientBoostingRegressor to estimate the residuals $\hat{r}(X)$.

