# OpenReview forum: "Epistemic Uncertainty Quantification To Improve Decisions From Black-Box Models"
_ICLR.cc/2026/Conference — ICLR 2026 Poster_

### Official Review · Reviewer_wB4E · 2025-10-31

**Soundness:** 3
**Presentation:** 3
**Contribution:** 3
**Rating:** 6
**Confidence:** 3

**Summary:**

This paper considers the excess risk, focussing on the grouping loss, of predictors. In particular, it proposes an estimator for these quantities in the context of proper scoring rules. The estimators are shown to have desirable properties such as consistency and unbiasedness, and instantiations with the 0-1 loss are considered. The method is evaluated on a semi-synthetic setting with known ground-truth probabilities and on a range of experiments involving LLMs and real data.

**Strengths:**

- The paper covers a valuable topic and does a good job of explaining exactly why considering the epistemic uncertainty quantification and, specifically, grouping loss, is important by including illustrative examples.
- The provided solution is reasonable, satisfies desirable properties, and performs well empirically.
- The paper reads well, and has clear notation and exposition for the formal results.
- There is a good range of empirical settings.

**Weaknesses:**

- The paper focuses on the binary setting, which is, of course, fairly limited.

There seems to be a leftover comment in the Appendix p. 13, l. 697.

**Questions:**

-

---

> ### Author Response · Authors · 2025-11-18
> **Authors's Response**
>
> We thank the reviewer for their time and feedback.
>
> > The paper focuses on the binary setting, which is, of course, fairly limited.
>
> It is true that the binary setting can be limited but **for multiclass cases, we can opt for top-label calibration**, a common calibration objective for multiclass problems since in many cases, there aren’t enough samples for class-wise calibration:
>
> **Definition:** For a $k$-class classification problem, a probabilistic classifier giving scores $f(X) = (f(X)_1, \ldots, f(X)_k)$ is **top-label-calibrated** if among all instances for which the confidence score of the predicted class is $p$, the probability that the predicted class is the correct one is $p$:
> $$P[Y = \arg\max(f(X)) \mid \max(f(X)) = p] = p.$$
>
>
> Top-label calibration is equivalent to simplifying the problem by reducing it to a binary task with output
>  $\tilde{Y}=1_{Y=\arg\max(f(X))}$. We would then define $c\circ f(X) = \mathbb{E}[1_{Y=\arg\max(f(X))}\mid \max(f(X))=p]\in[0,1]$. Then, our method still applies directly.
>
> This top-label formulation is closely related to controlling the probability of whether the answer given is correct or not, a framing used very frequently, for instance in LLMs.
>
> > There seems to be a leftover comment in the Appendix p. 13, l. 697.
>
> We apologize for the inconvenience, we erased it in the revised version we uploaded.

---

> > ### Comment · Reviewer_wB4E · 2025-11-25
> >
> > Thank you for your response.
> >
> > I appreciate the further clarification regarding the binary setting and agree with the authors' assessment about (the connection to) top-label calibration.
> >
> > I will keep my score.

---

### Official Review · Reviewer_SEdh · 2025-11-01

**Soundness:** 3
**Presentation:** 3
**Contribution:** 3
**Rating:** 8
**Confidence:** 2

**Summary:**

This paper introduces asymptotically consistent and sample-efficient lower-bound estimators for the grouping loss and excess risk, i.e., suboptimality of a prediction, serving as a complement to existing calibration metrics. The authors leverage the proposed estimator to design efficient LLM cascades that defer to stronger models, achieving higher accuracy at a lower cost than competing approaches.

**Strengths:**

1. The work paper is well structured, balancing the theoretical analysis and experimental validations.

2. Proofs for the related propositions are given in the appendix.

3. The extensive evaluations demonstrated the performance improvements convincingly.

**Weaknesses:**

1. Would the author be more explicit in indicating what the resources of the epistemic uncertainty are considered in the proposed method?

2. Minor: There is a lack of a REPRODUCIBILITY STATEMENT in the main body.

3. Minor: In the appendix, there is an unnecessary comment in green remaining. line 699.

**Questions:**

Please refer to the weakness.

---

> ### Author Response · Authors · 2025-11-18
> **Authors's Response**
>
> We thank the reviewer for their time and feedback.
> Thanks for noting points 2 and 3, we addressed them in the revised version we uploaded.
>
> > Would the author be more explicit in indicating what the resources of the epistemic uncertainty are considered in the proposed method?
>
> The epistemic loss arises because there is a discrepancy between the ground truth (unknown) posterior $P[Y|X]$, and the confidence scores $f(X)$ recovered from the LLM. Indeed, the LLM must answer Y= 0 or Y=1 given some information X based solely on its parametric knowledge, acquired through large-scale pretraining on a finite and potentially biased textual corpus. In other words, the model has not captured all statistical associations. Consequently, the model's output is not the ground truth posterior probability, but a learned statistical average derived from linguistic co-occurrence patterns. This discrepancy between the actual, real-world conditional probability and the LLM's learned internal approximation gives rise to  the epistemic loss.

---

> > ### Comment · Reviewer_SEdh · 2025-11-24
> > **Response**
> >
> > I thank the authors for their detailed answers. I'll keep my score, but please take into account that the LLM is quite far from my main research fields.

---

### Official Review · Reviewer_6jMk · 2025-11-01

**Soundness:** 3
**Presentation:** 3
**Contribution:** 4
**Rating:** 6
**Confidence:** 3

**Summary:**

The paper propose lower-bound estimators for grouping loss and excess risk, designed to address deficiencies in standard metrics (AUC, accuracy, calibration error) that conflate model confidence without properly distinguishing epistemic error or bias. Experiments cover semi-synthetic, tabular, and LLM question answering tasks, including cost-sensitive settings and cascades.

**Strengths:**

- The paper is generally well written. Extensive notational and methodological setup supports reproducibility and rigour.

- It advances epistemic uncertainty estimation using consistent, efficient lower-bound estimators,

- It addresses a critical gap for high-stakes AI applications.

**Weaknesses:**

- It does have dense mathematical sections which might make it harder to digest. Intuitive diagrams could help.

- Computational considerations for large-scale or real-time applications are missing. Scalability and latency constraints need more discussion.

- An ablation on tree depth is missing.

**Questions:**

- What are the practical limits for inference-time overhead with honest tree partitioning?

- How does the proposed approach adapt to free text tasks?

- How to select tree hyperparameter for new tasks or domains?

---

> ### Author Response · Authors · 2025-11-18
> **Authors's Response**
>
> We thank the reviewer for their time and feedback.
>
> > It does have dense mathematical sections which might make it harder to digest. Intuitive diagrams could help.
>
> We added a summary plot at the end of section 3.2 illustrating the risk estimator we propose. We improved the clarity and readability of the theoretical results in the main text. The updated sections are shown in red in the latest uploaded version.
>
> >  Computational considerations for large-scale or real-time applications are missing. Scalability and latency constraints need more discussion.
>
> We thank the reviewer for this comment. We report in the table below our measures of run time inferences for the models used. The honest tree’s inference time is measured when the prediction is made one individual sample at a time without parallelization to reflect a practical situation
>
> | Model       | Size                                 | Mean Inference Time (s/sample) |
> | ----------- | ------------------------------------ | ------------------------------ |
> | Honest Tree | max_depth= None, min_samples_leaf=15 | 0.00031 s                      |
> | Llama 3     | 1B                                   | 0.041 s                        |
> | Llama 3     | 3B                                   | 0.043 s                        |
> | Llama 3     | 8B                                   | 0.069 s                        |
> | Llama 3     | 70B                                  | 0.604 s                        |
> | Phi14       | 14B                                  | 0.060 s                        |
> | Gemma 2     | 27B                                  | 0.180 s                        |
> | Mixtral     | 8x7B                                 | 0.296 s                        |
>
>
> Experiments were run on a node equipped with 96 Intel Xeon Platinium 8468 CPU 2,10 GHz and one NVIDIA H100 GPU for the LLMs and on a node equipped with 40 Intel Xeon CPU E5-2698 v4 2.20GHz for the honest trees.
>
> Our tree is more than 100 times faster at inference time than the fastest LLM we consider (Llama 3 1B) and 2000 times faster than the slowest LLM (Llama 3 70B), showing that **our method adds a negligeable computational overhead**.
>
>  > An ablation on tree depth is missing.
>
> We added an ablation study on tree depth in Appendix F.4 and now refer to it in Section 3.3 by varying the min_samples_leaf parameter, which effectively controls tree depth. Overall, **our results are very robust to the min_samples_leaf value**. The results of the cascading experiments hold up to min_samples_leaf = 1,000 (which correspond to trees of depth around 6) and degrade at min_samples_leaf = 10,000 (trees of depth 3 approximately). This is expected as very coarse trees reduce the capacity of $\mathcal{R}^{GL}$ to improve over $\mathcal{R}^{CL}$ alone.
>
>
> > How does the proposed approach adapt to free text tasks?
>
> This is an excellent remark, as many complex questions cannot be naturally organized into an 'ordered' tabular format. Our proposed approach adopts a strategy similar to that used in Hu et al. [1]. We **embed the query using the all-MiniLM-L12-v2 model**—selected for its strong performance in Hu et al. [1]'s cascade experiments—to create the embedding space. In this resulting high-dimensional space, we fit a tree, as detailed in our additional experiments in Appendix F.2. When comparing our tree estimator to Hu et al. [1]’s baseline (which uses a k-nearest neighbors model fitted on the same embedding space with their optimized hyperparameters), **our tree-based approach consistently outperforms the baseline and the largest model of the cascade at a fraction of the cost**. While the high dimensionality of the embedding space somewhat reduces the overall gains, the results confirm that incorporating the grouping risk remains beneficial, as it allows for an improved accuracy compared to relying solely on the calibration risk.
>
> [1] RouterBench: A Benchmark for Multi-LLM Routing System, Hu et al. 2024
>
> > How to select tree hyperparameter for new tasks or domains?
>
> From our ablation study, the results are robust to the choice of tree depth. And our choice of keeping the min_sample_leaf at 15 across several tasks shows good results. If, in spite of this, results were not satisfactory on a new task, **tuning the min_sample_leaf hyperparameter would be the first thing to try**.

---

> > ### Author Response · Authors · 2025-11-28
> > **Rebuttal Discussion Period Ending Soon**
> >
> > Dear Reviewer,
> >
> > We appreciate your valuable time and input during the review process.
> >
> > With the discussion period ending soon, we wanted to ensure whether you had any remaining questions or required further clarification on our response.
> >
> > Thank you for your contributions.
> >
> > Best regards,
> >
> > The Authors

---

### Official Review · Reviewer_xMCH · 2025-11-04

**Soundness:** 1
**Presentation:** 1
**Contribution:** 1
**Rating:** 2
**Confidence:** 4

**Summary:**

The authors propose estimators of “grouping loss” and “excess risk”, and apply them to three decision problems: predicting the benefit of LLM finetuning, assessing LLM confidence, and routing queries between multiple LLMs.

**Strengths:**

Originality: the proposed estimators appear to be novel.

Quality: Figures 2, 4 and 6 communicate the authors’ results well.

Clarity: a good amount of the low-level text is easy to follow.

Significance: query routing is a potentially impactful application.

**Weaknesses:**

I believe the technical motivation of this work is unsound.

- First, describing an aleatoric-epistemic uncertainty decomposition as “crucial for reliable AI” and using that as the premise of the rest of the paper is not appropriate. The aleatoric-epistemic view has been shown to be incoherent, and subjective model-based uncertainties alone are not a legitimate basis for assessing model reliability (Bickford Smith et al, 2025).
- Second, dismissing standard evaluation metrics—particularly proper scoring rules, which are very well established and have strong theoretical foundations (Savage, 1971)—in favour of new quantities ought to be extremely well argued from a technical perspective, yet it isn’t. Instead the authors offer an assortment of statements about various metrics, then quickly land on “grouping loss” and “excess risk” as the quantities of apparent interest.

Even if we were to set this aside, the practical contribution is hard to get behind.

- In Section 4.2 we see results for predicting the benefit of LLM finetuning. The practical significance of these is unclear. We see a scatter plot of the “excess risk” estimator against “cost reduction”, but the magnitudes of these quantities are hard to interpret from the plot. This is a nonstandard setup that needs explaining better or supplementing with a more practically relatable setup. It also appears a basic baseline method is missing: why not just evaluate the “cost” on the training set, if this is what we care about?
- In Section 4.3 there isn’t a well-defined task or notion of success. Instead the authors just report confidence results (confidence is not a quantity they are proposing) in Figure 1 and “grouping loss” results in Figure 4. What are we to do with these?
- In Section 4.4 the focus is on routing a query to an appropriate LLM, balancing prediction costs against predictive accuracy. The results in Figure 5 appear to be positive, although the choice to stop at $t=0.001$ gives a false impression of how expensive the proposed method can be. This high cost is clear from Figure 6 (right): even with the baseline method in a high-cost configuration ($\lambda=100$), the proposed method is up to ~60 percent more expensive, and the payoff in terms of accuracy is small, at less than ~3 percentage points. At a higher level it seems the extent to which the results are positive here is highly contingent on the dataset having many examples that can be accurately classified by the smaller models.

---

Bickford Smith et al (2025). Rethinking aleatoric and epistemic uncertainty. ICML.

Savage (1971). Elicitation of personal probabilities and expectations. Journal of the American Statistical Association.

**Questions:**

Can you provide a technical argument for the insufficiency of proper scoring rules? Note that any appeal to the idea of epistemic/reducible uncertainty would have to be rigorously derived, not just stated as important.

Section 4.2:

- Can you clarify the cost matrices you used?
- Can you explain the axis magnitudes in Figure 3?
- Why not just evaluate the “cost” on the training set, if this is what we care about?

Section 4.3: what is the concrete task or notion of success you have in mind?

Section 4.4: can you extend the lines in Figure 5 to include $t=0$?

---

> ### Author Response · Authors · 2025-11-19
> **Authors's Main Response**
>
> We thank the reviewer for their feedback and provide below detailed answers.
>
>
> > The aleatoric-epistemic view has been shown to be incoherent [...] (Bickford Smith et al, 2025)
>
> We thank the reviewer for this reference, which reveals the multiplicity of mathematical definitions of aleatoric and epistemic uncertainties.
>
> The reference argues that the quantities defined solely based on the learned distribution $p_n$ aren’t a legitimate basis for assessing a model’s reliability “because we expect $p_n(z)$ to be imperfect” (section 3.7). It therefore claims that "Some kind of external grounding is crucial for well-informed practical deployment" (end of p 7). **This is exactly what we propose**. Our epistemic loss (eq. 1) is defined with regards to an external grounding $f^*(X)$, and our estimator confronts the learned values $f(X)$ with ground-truth realization ($Y$ in our case).
>
> **Our work aligns with Bickford 2025 on multiple points of their conclusion**: point d) on using external grounding, detailed above, but also point a) advocating for a metric adapted to decision-making (section 2.3 and 3.2 of our paper).
>
> We have added Bickford Smith 2025 to our related works (around line 201) in the revised version uploaded.
>
> >“dismissing standard evaluation metrics—particularly proper scoring rules [...] in favour of new quantities ought to be extremely well argued from a technical perspective, yet it isn’t.”
>
> Thank you for pointing this out. We are happy to revise the text to make it clear that we build upon, rather than dismiss, proper scoring rules. But proper scoring rules alone are insufficient, e.g. in our application on deciding when to defer a query, for two main reasons:
> - **Locality:** Proper scoring rules hold on average, while our estimator provides **local, per-input error** allowing to make individual routing decisions.
> - **In the presence of aleatoric uncertainty, the highest possible accuracy (or lowest possible proper scoring rule) is unknown**. Proper scoring rules or accuracy measure absolute performance. When the best possible score is unknown due to aleatoric uncertainty, this is a poor basis for deferral. **Our quantities instead measure the "room for improvement"** which is best suited to decide whether to defer or not.
>
> > In Section 4.2 why not just evaluate the “cost” on the training set
>
> This experiment serves as a sanity check on real data, confirming that our estimator effectively captures the excess-risk. Here, we could indeed simply evaluate the cost on the train set, but such a quantity only provides a dataset-level measure, whereas our estimator provides sample-level measures that are crucial for applications like LLM routing or fairness audits.
>
> > Section 4.3 what is the concrete task or notion of success you have in mind?
>
> Section 4.3 does not introduce a notion of success. Instead, it demonstrates the existence of grouping loss in LLMs as a way to motivate future research directions. For instance, the observation that neural networks (and later LLMs [2]) exhibit systematic miscalibration triggered a large line of work (Guo et al. [1] cited more than 8 000 times). **Even without a formal measure of success, high miscalibration has been highly effective in stimulating a whole research field to improve prediction uncertainty**.
>
> [1] On the calibration of modern neural networks, Guo et al. ICML 2017
>
> [2] Evaluating language models as risk scores, Cruz et al. NeurIPS 2024
>
> > In Section 4.4 the [...] high cost is clear from Figure 6 (right): [...] the proposed method is up to ~60 percent more expensive, and the payoff in terms of accuracy is small
>
> A new version of figure 5 including the $t=0$ case has been integrated in the revised manuscript . Overall, **including $t=0$ leaves the final results almost unchanged**, because the risk values estimated are sufficiently high to have already been detected with $t=0.01$.
>
> While the right subfigure of Figure 6 shows that in extreme cases, our method's cost can be up to 60% higher relative to the predictive router baseline, it is on average only 6\% costlier. The fact that **our method also systematically improves accuracy by up to 3 points** (which compared to improvements typically reported in papers is relatively high) compared to this baseline makes it a more competitive alternative.
>
> > it seems the extent to which the results are positive here is highly contingent on the dataset having many examples that can be accurately classified by the smaller models.
>
> Our method's performance gain is proportional to the small model's accuracy, as the better the small model can correctly predict a larger subset of instances, the larger the performance gains we measure. However, even with small models of modest accuracy (e.g., 39% (Llama 3 3B) or 64% (Llama 3 1B, Llama 3 8B) compared to 77% for the best model (Mixtral8x7B)), our approach **still significantly improves performance on top of the best and largest model overall** (Fig 5).

---

> ### Author Response · Authors · 2025-11-19
> **Authors's Clarification of Experiments**
>
> >the authors just report confidence results [...] in Figure 1 [...] What are we to do with these?
>
> Figure 1 illustrates the interpretability of our method in the case where the attributes are sensitive. The regions created by the honest tree allow to have clear definitions of the groups for which the evaluated model shows under- or overconfidence.
>
> > Section 4.2:
> > Can you clarify the cost matrices you used?
>
> We thank the reviewer for this clarification, we added it in the appendix D.2.
>
> We define **binary cost matrices $\Lambda \in \mathbb{R}^{2 \times 2}$** of the shape:
>
> $$\Lambda =
> \begin{bmatrix}
>  -1 & 0 \\\\
>  0 & \Lambda_{11}
> \end{bmatrix}$$
>
> with $\Lambda_{11} \in \mathbb{R}^{-}$.
>
> We select values of $\Lambda_{11}$ so that the **optimal threshold $t^*$** takes values $[0.01, 0.025, 0.05, 0.1, 0.25, 0.5, 0.75, 0.9, 0.95, 0.975, 0.99]$.
>
>
> > Can you explain the axis magnitudes in Figure 3?
>
> In our setup, the magnitude of the risk is given by $\Lambda_{\Delta}$, as can be seen in equation 6. Therefore the axis magnitudes represent the normalized magnitudes of the risks for each setup of cost matrices. (Each cost matrix induces a different magnitude of risk, thus for the benefits to be more easily comparable, we normalize them by the risk coefficient $\Lambda_{\Delta}$).

---

> > ### Author Response · Authors · 2025-11-28
> > **Rebuttal Discussion Period Ending Soon**
> >
> > Dear Reviewer,
> >
> > We appreciate your valuable time and input during the review process.
> >
> > With the discussion period ending soon, we wanted to ensure whether you had any remaining questions or required further clarification on our response.
> >
> > Thank you for your contributions.
> >
> > Best regards,
> >
> > The Authors

---

### Author Response · Authors · 2025-12-03
**Summary of the discussions**

Dear reviewers, dear AC,

We thank you for your time to review our work. We summarize the exchanges with the reviewers in this comment.

## Context
Our work proposes measures of epistemic error and epistemic risk in a decision making setting.

## Reviewer 1 (xMCH):
The reviewer seemed to misread the theoretical setting of our work, resulting in an outlying low score compared to the other reviews. The reviewer cited a paper [1], stressing that definitions of the epistemic uncertainty abound, but few have desirable properties. Our definition (eq.1 of our paper) relies on the specific properties put forward in [1], and our work is aligned with [1]’s concluding points a) and d). The reviewer thought that we discarded proper scoring rules. Rather, we build on them (thm. 1 of our paper), estimating the “pointwise room for improvement” of models, which is not possible with existing metrics, and unlock our application on LLM query routing.

As requested by reviewer xMCH, we clarified the specificities of our setup and developed the results corresponding to the measure of grouping loss for LLMs, drawing a parallel to the results of famous papers that spurred the research field [2].
The reviewer feared that our proposed method could be too expensive for what they deemed  a “small” gain (3 accuracy points). We argue that their concern on cost is based on an extreme point where our method is 60% relatively more expensive to the baseline whereas on average, our method is marginally (6%) costlier than the baseline while **systematically improving accuracy by up to 3 points** (which we argue is high compared to results typically reported).

[1] Rethinking aleatoric and epistemic uncertainty, Bickford Smith et al. ICML 2025

[2] On the calibration of modern neural networks, Guo et al. ICML 2017

## Reviewer 2 (6jMk):
The reviewer was enthusiastic about the paper’s contribution, whether theoretical or experimental, and appreciated the writing and rigorous “extensive notational and methodological setup”.
- They found the mathematical sections “dense” making the paper “harder to digest”. We rewrote these parts in more detail and added a new diagram (fig. 7) to better understand our proposed risk estimator.
- They raised the lack of ablation study on our method’s depth (honest trees based). We added it in Appendix F.4, showing that **our results are very robust to the min_samples_leaf value and therefore robust down to a depth of around 6**.
- They asked for the free-text generalization of our method and we referred to Appendix F.2, where we had embedded the query using an embedding model. In this section, while the high dimensionality of the embedding space somewhat reduces the overall gains, **our tree-based approach consistently outperforms the baseline and the largest model of the cascade at a fraction of the cost**.
- They wondered whether the proposed method induced an inference-time overhead. We showed that **our method adds a negligeable computational overhead**, being 100 times faster on average than the fastest LLM considered.

## Reviewer 3 (SEdh):
The reviewer appreciated the structure of our work, especially the balance between theoretical and experimental results, noting that “the extensive evaluations demonstrated the performance improvements convincingly.” They requested clarification about the “sources of the epistemic uncertainty […] considered in the proposed methods”. In our response, we detailed the sources of epistemic uncertainty when evaluating LLMs, and the reviewer subsequently confirmed their score.

## Reviewer 4 (wB4E):
The reviewer found that our proposed estimator “performs well empirically” on “a good range of empirical settings” and valued our work “explaining exactly why considering the epistemic uncertainty quantification and, specifically, grouping loss, is important by including illustrative examples”. The only weakness raised was an inquiry about the multiclass generalization of our work. We proposed using top-label calibration to reduce the problem to a binary setting and explained how this formulation connects to our framework. The reviewer confirmed their score.

---

### Meta-Review · Area_Chair_vV4c · 2026-01-06

**Summary:**

Most reviewers found the theoretical development sound and appreciated the careful formalization of grouping loss and excess risk, as well as the extensive empirical evaluation, particularly for LLM cascades and cost-sensitive decision making. The rebuttal successfully addressed many technical and experimental concerns by adding ablations, clarifying computational overhead, improving exposition, and strengthening the connection to practical LLM routing scenarios. However, one reviewer raised a fundamental objection to the epistemic uncertainty framing and questioned whether the proposed metrics offer a sufficiently well-justified and practically necessary alternative to established evaluation tools such as proper scoring rules.
While the authors provided thoughtful clarifications and carefully positioned their work relative to recent critiques, some degree of conceptual disagreement regarding the framing of epistemic uncertainty remains. Nevertheless, considering the overall strength of the theoretical development　and quality of the empirical evaluation, and the positive feedback from three reviewers, I believe the contribution meets the bar for acceptance. The rebuttal addressed many technical and experimental concerns, and I strongly encourage the authors to fully incorporate these improvements in the final version.  Accordingly, I recommend acceptance.

**Reviewer Concerns:**

Reviewer xMCH:
The rebuttal clarified that the proposed epistemic risk and grouping loss build upon proper scoring rules rather than dismissing them, and that the estimators rely on external grounding in line with recent critiques of epistemic uncertainty. However, the reviewer’s fundamental concern about the soundness of the epistemic–aleatoric framing and skepticism about the practical necessity and interpretability of the proposed quantities remain vastly outstanding.

Reviewer 6jMk:
Concerns regarding mathematical density, lack of ablation on tree depth, inference-time overhead, and applicability to free-text tasks were convincingly addressed through additional explanations, new diagrams, ablation studies, runtime measurements, and embedding-based extensions. No major outstanding concerns remain from this reviewer.

Reviewer SEdh:
Requests for clarification on the sources of epistemic uncertainty and minor presentation issues were addressed in the rebuttal and revised manuscript. The reviewer explicitly indicated satisfaction and maintained their score.

Reviewer wB4E:
The concern about the restriction to binary classification was addressed through a clear explanation of top-label calibration and its connection to the proposed framework, and minor presentation issues were fixed. No substantive concerns remain outstanding.

**Reviewer Scores:**

Reviewer xMCH:
The score would likely remain unchanged, as the rebuttal clarified the authors’ alignment with proper scoring rules and external grounding, but did not fully resolve the reviewer’s fundamental disagreement with the epistemic–aleatoric framing and the perceived practical value of the proposed metrics.

Reviewer 6jMk:
The score would remain unchanged, since the rebuttal addressed concerns about ablations, scalability, and inference-time overhead without altering the reviewer’s already positive overall assessment.

Reviewer SEdh:
The score would remain unchanged, as the reviewer’s minor concerns regarding sources of epistemic uncertainty and presentation details were satisfactorily addressed in the rebuttal.

Reviewer wB4E:
The score would remain unchanged, given that the clarification on the binary setting and its extension via top-label calibration resolved the reviewer’s main concern.

---

### Decision · Program_Chairs · 2026-01-26

Accept (Poster)